

# Effects of nitrogen deposition on growing-season soil methane sink across global forest biomes

Enzai Du[1,2*], Nan Xia[2], Wim de Vries[3,4]

[1]State Key Laboratory of Earth Surface Processes and Resource Ecology, Faculty of Geographical Science, Beijing Normal University, Beijing100875, China

[2]School of Natural Resources, Faculty of Geographical Science, Beijing Normal University, Beijing100875, China

[3]Wageningen University and Research, Environmental Research, PO Box 47, NL-6700 AA Wageningen, the Netherlands

[4]Wageningen University and Research, Environmental Systems Analysis Group, PO Box 47, NL-6700 AA Wageningen, the Netherlands

*Correspondence to*: Enzai Du (enzaidu@bnu.edu.cn)

**Abstract.** Anthropogenic alteration of global nitrogen (N) deposition has resulted in profound impacts on soil fluxes of greenhouse gases in terrestrial ecosystems. However, the response of soil methane ($CH_4$) flux to N deposition remains poorly quantified in global forest. Based on a synthesis of experimental results from literature, we evaluated the effects of N deposition on growing-season soil $CH_4$ flux across forest biomes. A distinction was made between low-level N addition that is comparable with the worldwide range in N deposition ($< 60$ kg $N^{-1}$ $yr^{-1}$) and high-level N addition ($> 60$ kg $N^{-1}$ $yr^{-1}$). The results showed that growing-season soil $CH_4$ flux was significantly affected by N additions, the value being dependent on the N addition level and forest biome. Low-level N addition significantly increased growing-season soil $CH_4$ uptake in boreal forest, while an opposite effect occurred in temperate and subtropical forests. However, high-level N addition significantly decreased growing-season soil $CH_4$ uptake across boreal, temperate, and subtropical forests. At biome scale, current N deposition was estimate to increase growing-season soil $CH_4$ sink by 0.029 Tg $CH_4$ in boreal forest, while it decreased growing-season soil $CH_4$ sink by 0.025 Tg $CH_4$ and 0.051 Tg $CH_4$ in temperate and subtropical forests, respectively. This work improves our understanding of biome-specific effect of N deposition on soil $CH_4$ uptake and identifies knowledge gaps in the effect of N deposition on soil $CH_4$ flux in tropical forest.

## 1 Introduction

Atmosphere methane ($CH_4$) contributes significantly to climate warming via a greenhouse effect and has shown an increasing concentration over recent decades (Dlugokencky et al., 2011; Kirschke et al., 2013). Forest soils are an important sink of atmospheric $CH_4$ that dominates the total soil $CH_4$ sink in terrestrial ecosystems (Le Mer and Roger, 2001; Dutaur and Verchot, 2007). Anthropogenic nitrogen (N) emissions have enhanced global N deposition (Galloway et al. 2003 & 2008) and caused profound impacts on forest ecosystems, including a stimulation on net primary productivity in N-limited forests (De Vires et al., 2014 & 2017; Du and De Vries, 2018; Schulte-Uebbing and De Vries, 2018) and many adverse effects via excess N inputs, such as soil acidification, nutrient imbalances, biodiversity loss and even a reduction in forest growth (Aber



et al., 1998; Bowman et al., 2008; Bobbink et al., 2010; Du et al., 2016). However, the response of soil $CH_4$ flux to N deposition remains poorly quantified across global forest biomes and is usually ignored in previous assessments of global greenhouse gas emissions (Bodelier and Steenbergh, 2014; De Vries et al., 2017).

Soil $CH_4$ sink results from a higher rate of $CH_4$ oxidation by methanotrophs than that of methanogenesis by methanogens, both of which are sensitive to external N additions (Schnell and King, 1994; Le Mer and Roger, 2001; Bodelier and Laanbroek, 2004). In a N-deficient condition, low-level N inputs are expected to release the N limitation of methanotrophic microorganisms and/or the biosynthesis of enzymes involved in methane oxidation and thus increase soil $CH_4$ uptake (Bodelier et al., 2000; Bodelier and Laanbroek, 2004; Reay and Nedwell, 2004). Nitrogen addition may also increase in the size and activity of the nitrifying population, co-oxidizing atmospheric $CH_4$ (Reay et al., 2005). In contrast, high-level N inputs can decrease soil $CH_4$ uptake via (i) a direct inhibition of $CH_4$ oxidation due to either osmotic stress caused by increased soil inorganic N concentrations or competitive inhibition of the enzyme methane mono-oxygenase by ammonia (Schnell and King, 1994; Sitaula et al., 1995; Bodelier and Laanbroek, 2004) and (ii) an indirect effect due to soil acidification and an imbalance of N and P (Veraart et al., 2015). Moreover, external N inputs can stimulate forest growth (Sonnleitner et al., 2001; Schulte-Uebbing and De Vries, 2018) and increase evapotranspiration, indirectly favouring microbial $CH_4$ oxidation in soils.

Based on a meta-analysis of N addition experiments, Liu and Greaver (2009) showed that forest soil $CH_4$ uptake was significantly reduced by N additions. However, another meta-analysis indicated an overall neutral effect of N additions on forest soil $CH_4$ uptake (Aronson and Helliker, 2010). This inconsistency may be caused by not distinguishing between the N limitation of forests and the N addition level, both likely affecting the impacts of N additions on soil $CH_4$ sink in forest. Previous studies have indicated a general trend of decreasing N limitation from geologically young boreal forests towards tropical forests (Elser et al. 2007; Hedin et al., 2009; Vitousek et al., 2010; Du and De Vries, 2018). In view of an increase in N availability from boreal to tropical forests, the response of soil $CH_4$ flux to N addition is expected to vary across forest biomes. In line with the mechanisms described above, we hypothesize that an increase in N availability by N deposition increases the $CH_4$ sink in boreal forest but reduces that sink in temperate, and (sub) tropical forests. This potential biome-specific effect of N deposition has not been well described yet.

As shown in a recent assessment conducted by the World Meteorological Organization (WMO) Global Atmosphere Watch programme (GAW), N deposition in eastern and southern China represents a maximum level among the hotspots of N deposition across the globe (Vet et al., 2014). Nitrogen deposition is estimated to be on average 22 kg N ha$^{-1}$ yr$^{-1}$ in China's forest, but in distinct regions values can be much higher although only very occasionally higher than 60 kg N ha$^{-1}$ yr$^{-1}$ (Du et al., 2014a & 2016). A dose of 60 kg N ha$^{-1}$ yr$^{-1}$ can thus be used as a threshold for the maximum N deposition on the globe. However, higher N dosages (e.g., > 100 kg N ha$^{-1}$ yr$^{-1}$) have been frequently applied by existing N addition experiments (Liu and Greaver, 2009; Aronson and Helliker, 2010; Table S1). Based on a synthesis of experimental results in literature, here



we evaluated the effects of low-level N addition ($< 60$ kg $N^{-1}$ $yr^{-1}$, being in the worldwide range of N deposition) and high-level N addition ($> 60$ kg $N^{-1}$ $yr^{-1}$) on growing-season soil $CH_4$ flux across global forest biomes. Our analysis focused on growing-season soil $CH_4$ sink because it accounts for a major proportion of the annual $CH_4$ sink (Le Mer and Roger, 2001) and the reported data in literature were usually measured during the growing season. Moreover, we estimated the biome-scale effect of N deposition on growing-season soil $CH_4$ sink based on the response ratio to low-level N addition, mean forest-specific N deposition, the length of growing season and the area of each forest biome.

## 2 Data and Method

### 2.1 Data set

By conducting a survey of the online library of ISI Web of Science (http://isiknowledge.com), Google Scholar (https://scholar.google.com) and China National Knowledge Infrastructure (http://www.cnki.net/), we collected experimental data on the effects of N additions on growing-season soil $CH_4$ flux in forest ecosystems across the globe. The key words 'methane (or $CH_4$)', 'nitrogen addition (or nitrogen fertilization, nitrogen deposition)' and 'forest' were used. We recorded data on the growing-season soil $CH_4$ flux in control plots and treatment plots, as well as information on N addition rates, N forms, site location (latitude and longitude), forest type, mean annual temperature (MAT), mean annual precipitation (MAP), growing-season mean temperature (GSMT), and growing-season mean monthly precipitation (GSMP). Soil $CH_4$ flux was measured using closed static chamber methodology and analyzed using gas chromatography. A negative value of soil $CH_4$ flux indicates an uptake of $CH_4$ by soils, while a positive value indicates an emission of $CH_4$ from soils. Our database only included data on N addition experiments by using $NH_4NO_3$ as well as nitrate (e.g., $NaNO_3$) and ammonium (e.g., $NH_4Cl$) based N forms. We excluded experiments by applying urea and manure, because these organic N forms have limited implications for the effects of N deposition (Aronson and Helliker, 2010) as N deposition mainly occurs in forms of inorganic N (Vet et al., 2014).

Overall, our database included experimental results of 25 forests across 18 sites in northern hemisphere (Fig. 1). More specifically, 22 experiments were conducted by using $NH_4NO_3$ in 22 forests and six experiments by using nitrate ($NaNO_3$ or $KNO_3$) and ammonium ($NH_4Cl$) based N forms in three forests (Table S1). For the latter three forests, the effect of nitrate and ammonium based N additions were averaged for each N dosage and used as one treatment for further analysis. We did so to harmonize the different effects of reduced and oxidized N forms and thus make the results comparable with those based on $NH_4NO_3$. In summary, our database included data on eight N treatments for boreal forests, 22 N treatments for temperate forest and 17 N treatments for subtropical forest, respectively (Table S1). Unfortunately, we found no report on the effect of inorganic N additions on soil $CH_4$ flux in tropical forest. We thus were not able to assess the effect of N deposition on growing-season soil $CH_4$ flux in tropical forest and compare it with other forest biomes.



## 2.2 Statistical analysis

We calculated a response ratio ($R_{CH4-N}$, g $CH_4$ $kg^{-1}$ N) of growing-season soil $CH_4$ flux to each N treatment according to Eq. 1,

$$R_{CH_4-N} = \frac{Flux_{treatment} - Flux_{control}}{N_{add}} \times 87600 \quad (1)$$

where $N_{add}$ indicates the N addition for each treatment (kg N $ha^{-1}$ $yr^{-1}$), $Flux_{treatment}$ and $Flux_{control}$ indicate the mean growing-season soil $CH_4$ flux (mg $CH_4$ $m^{-2}$ $h^{-1}$) in the treatment plots and control plots, respectively, and the constant 87600 is a unit correction factor from mg $m^{-2}$ $h^{-1}$ to g $CH_4$ $ha^{-1}$ $yr^{-1}$. A positive value of the response ratio indicates a reduction in soil $CH_4$ uptake or an increase in soil $CH_4$ emission due to N addition, while a negative value indicate an opposite effect.

We conducted a Student's t-test to compare growing-season soil $CH_4$ fluxes in control plots and their response ratios to low-
level N addition (< 60 kg $N^{-1}$ $ha^{-1}$ $yr^{-1}$) and high-level N addition (> 60 kg $N^{-1}$ $ha^{-1}$ $yr^{-1}$) across boreal, temperate and subtropical forests, respectively. The low-level N addition is in the range of N deposition over the world (Vet et al., 2014). A multiple linear regression model was used to evaluate the role of GSMT, GSMP, N addition and their interactions in regulating the response ratio of growing-season soil $CH_4$ flux to N addition. Moreover, biome-scale effects of N deposition on growing-season soil $CH_4$ sink were estimated based on the response ratio of growing-season soil $CH_4$ flux to low-level N
addition (< 60 kg $N^{-1}$ $ha^{-1}$ $yr^{-1}$), mean forest-specific total N deposition (base year 2010) (Schwede et al., 2018), the length of growing season and the area of each biome (base year 2010) (Keenan et al., 2015). We used average growing season lengths of 4 months, 7 months and 12 months for boreal, temperate and subtropical forests, respectively (Piao et al., 2007). Values were mean ± standard error, if not specially noted. All statistical analysis was performed using R software (version 3.4.0; R Development Core Team, 2017; http://www.r-project.org/) with a significance level of p < 0.05.

## 3 Results and discussion

### 3.1 Variations in growing-season soil $CH_4$ uptake across forest biomes

Forest soils showed a significant uptake of $CH_4$ during growing season in control plots, on average being -0.281±0.120, -0.050±0.016, and -0.023±0.006 mg $CH_4$ $m^{-2}$ $h^{-1}$ in boreal, temperate and subtropical forests, respectively (Table 1). Specifically, the rate of soil $CH_4$ uptake was significantly higher in boreal forest than temperate (*p*=0.01) and subtropical
forests (*p*=0.004). Moreover, the rate of growing-season soil $CH_4$ uptake in temperate forest showed no significant difference from that in subtropical forest (*p*=0.12), even though its mean values were twice as high (Table 1). Considering the difference in the lengths of growing season, we estimated that the total $CH_4$ uptake per growing season was -8.2±3.5, -2.6 ±0.8, and -2.0±0.5 kg $CH_4$ $ha^{-1}$ boreal, temperate and subtropical forests, respectively. Compared with boreal forest, the lower rates of soil $CH_4$ uptake in temperate and subtropical forests might be attributed to a suppression of $CH_4$ oxidation by



more growing-season precipitation (Whittaker, 1962; Le Mer and Roger, 2001), higher background N availability (Vet et al., 2014; Schwede et al., 2018; Du and De Vries, 2018) and a stronger phosphorus limitation to methanotrophic microorganisms (Veraart et al., 2015).

By multiplying the average rate of soil $CH_4$ uptake in control plots with the length of growing season and the area of each
forest biome (Keenan et al., 2015), we estimated that growing-season soil $CH_4$ sinks were -10.13±4.33, -1.74±0.56, and -0.64±0.17 Tg $CH_4$ in global boreal, temperate, and subtropical forest biomes, respectively (Table 1). Based on the mean rate of growing-season soil $CH_4$ uptake (-0.038±0.029 mg $CH_4$ $m^{-2}$ $h^{-1}$) for tropical forest (Dutaur and Verchot, 2007), biome-scale soil $CH_4$ sink in tropical forest was estimated to be -5.99±4.57 Tg $CH_4$. Overall, global forest soils contributed to a $CH_4$ sink of 18.5 Tg $CH_4$ during growing season, accounting for more than half of the annual sum soil $CH_4$ sink (~30 Tg $CH_4$ $yr^{-1}$)
in global terrestrial biomes (Kirschke et al., 2013; Murguia-Flores et al., 2018). If the amount of soil $CH_4$ sink in non-growing season is accounted, forest soils can contribute an even higher proportion of global soil $CH_4$ sinks. Our analysis indicates that forest biomes dominate terrestrial soil $CH_4$ sinks at global scale.

### 3.2 Responses of growing-season soil $CH_4$ uptake to nitrogen additions

The response ratio of growing-season soil $CH_4$ flux to N addition varied significantly with N dosage ($p=0.001$, Table 2). In
line with our hypothesis, the effect of low-level N addition (< 60 kg N $ha^{-1}$ $yr^{-1}$) on growing-season soil $CH_4$ flux was different from that of high-level N addition (> 60 kg N $ha^{-1}$ $yr^{-1}$) for each forest biome (Fig. 2). Specifically, low-level N addition significantly increased growing-season soil $CH_4$ uptake in boreal forest (-59.3±28.6 g $CH_4$ $kg^{-1}$ N, $p<0.05$), while high-level N addition significantly decreased it (23.1±9.4 g $CH_4$ $kg^{-1}$ N, $p<0.05$). In temperate forest, growing-season soil $CH_4$ uptake was significantly decreased by both low-level N addition (8.9±5.5 g $CH_4$ $kg^{-1}$ N, $p<0.05$) and high-level N
addition (15.2±8.8 g $CH_4$ $kg^{-1}$ N, $p<0.05$), while the effect of low-level N addition showed no significant difference from that of high-level N addition ($p=0.26$). Similarly, soil $CH_4$ uptake in subtropical forest was significantly decreased by both low-level N addition (10.8±3.1 g $CH_4$ $kg^{-1}$ N, $p < 0.01$) and high-level N addition (5.5±2.1 g $CH_4$ $kg^{-1}$ N, $p< 0.01$), and the effect of low-level N addition showed a marginal difference from that by high-level N addition ($p=0.08$). Our results indicate the necessity to separately assess the effect of low-level N addition from that of high-level N addition and distinguish between
forest biomes. It supports our hypothesis that in boreal forest N deposition enhances the soil $CH_4$ sink whereas the reverse is true for temperate and subtropical forests.

The response ratio of growing-season soil $CH_4$ flux to N addition varied significantly with growing-season mean temperature ($p=0.017$, Table 2), reflecting a variation in the effect of N addition across forest biomes. For instance, soil $CH_4$ uptake was increased by low-level N addition in boreal forest and the response ratio was significantly different from those in temperate
($p<0.001$) and subtropical forests ($p=0.008$), while no significant difference was found between the latter two ($p=0.41$). The response ratio to high-level N addition in boreal forest was not significantly different from that in temperate forest ($p=0.32$),





while it is significantly higher than that in subtropical forest ($p$=0.007). As boreal forest is characterized by a strong N limitation (Du and De Vries, 2018), the increase of soil $CH_4$ uptake by low-level N addition was likely due to releasing N limitation of methanotrophic microorganisms (Bodelier et al., 2000; Bodelier and Laanbroek, 2004). In temperate and subtropical forests, the reduction of soil $CH_4$ uptake by simulated N deposition is likely due to a direct inhibition of $CH_4$

oxidation by increased inorganic N concentrations (Schnell and King, 1994; Sitaula et al., 1995; Bodelier and Laanbroek, 2004) and/or an indirect effect due to soil acidification and an imbalance of N and P (Veraart et al., 2015).

### 3.3 Biome-scale effect of nitrogen deposition on growing-season soil $CH_4$ sink

Based on the average response ratio of growing-season soil $CH_4$ flux to simulated N deposition, the average N deposition (Schwede et al., 2018), the length of growing season and the area of each biome (Keenan et al., 2015), we roughly estimated

the effect of N deposition on growing-season soil $CH_4$ sink in global forest biomes. At a biome scale, N deposition increased growing-season soil $CH_4$ sink by 0.029 Tg $CH_4$ in boreal forest, while it decreased growing-season soil $CH_4$ sink by 0.025 Tg $CH_4$ and 0.051 Tg $CH_4$ in temperate and subtropical forests, respectively (Table 1). However, a large uncertainty remains in the effect of N deposition on soil $CH_4$ sink in tropical forest, especially considering that tropical forest (1798 million ha) accounts for 45% of the global forest area (Keenan et al., 2015). By using a same response ratio as for subtropical forest

($10.08 \pm 3.1$ g $CH_4$ $kg^{-1}$ N), we estimated that N deposition in tropical forest (on average 7.2 kg N $ha^{-1}$ $yr^{-1}$, Schwede et al., 2018) decreased soil $CH_4$ sink by $0.14 \pm 0.04$ Tg $CH_4$ at the biome scale. Overall, N deposition likely decreased growing-season soil $CH_4$ sink by 0.187 Tg $CH_4$ in global forest. However, this reduction is negligible compared with the total growing-season soil $CH_4$ sink in global forest (-18.5 Tg $CH_4$).

Previous meta-analyses, not differentiating the effect of low-level N addition from high-level N addition, nor between

biomes (Liu and Greaver, 2009; Aronson and Helliker, 2010), have likely overestimated the negative effect of N deposition on forest soil $CH_4$ uptake at global scale. Using the constant response ratio of Liu and Greaver (2009) of $16 \pm 4$ g $CH_4$ $kg^{-1}$ N for forests and multiplying with the global mean N deposition (6.0 kg N $ha^{-1}$ $yr^{-1}$, Schwede et al., 2018) and global total forest (4016 million ha, Keenan et al., 2015) gives an reduction of $CH_4$ sink by 0.386 Tg $CH_4$ $yr^{-1}$, whereas we estimate a much lower reduction by 0.187 Tg $CH_4$ $yr^{-1}$. Considering that $CH_4$ is 25 times more effective, on a per-unit-mass basis, than

$CO_2$ in absorbing long-wave radiation on a 100 year time horizon (Forster, et al., 2007), we further estimate that the warming effect of N deposition induced reduction of growing-season soil $CH_4$ sink in global forest is equivalent to an emission of 4.68 Tg $CO_2$. This effect on climate warming is negligible in view of a large carbon sink (2.4 Pg C $yr^{-1}$) in global established forest (Pan et al., 2011).

### 3.4 Uncertainties and implications

Based on an analysis of the results from existing N addition experiments, we first assessed the different effects of low-level N addition on soil $CH_4$ uptake from that of high-level N addition and then estimated the effect of N deposition on soil $CH_4$





sink across global forest biomes. Compared with the two existing meta-analyses (Liu and Greaver, 2009; Aronson and Helliker, 2010), the results have improved our understanding of the effect of N deposition on soil $CH_4$ sink in different forest biomes. However, uncertainties remain due to relatively limited number of manipulated experiments by using low-level dosage of inorganic N addition, especially in tropical forests. To date, only one experimental study has been published on the

effect of urea addition on soil $CH_4$ flux in tropical forest (Veldkamp et al., 2013), but the results based on N additions by using urea shed limited lights on the effect of N deposition. As N deposition in many tropical regions is expected to increase in the future (Lamarque et al., 2013), further efforts are needed to evaluate the effect of N deposition on soil $CH_4$ flux in tropical forest by conducting extra N addition experiments with low-level additions of inorganic N. Moreover, the existing experimental studies are located unevenly in space. For instance, our search of the literature found no experimental reports

on effects of N deposition on soil $CH_4$ flux in southern hemisphere. Experimental results of subtropical forests are mainly from Asia, while no such studies have been reported in Africa and South America with a large extent of subtropical forest.

A recent literature review indicates that forest soil $CH_4$ sink in northern hemisphere has significantly declined during past three decades and this trend has been attributed to an increase of precipitation (Ni and Groffman, 2018). Our analysis indicates that increasing N deposition in developing countries (e.g., China and India) (Liu et al., 2013; Abrol et al. 2017)

may partially contribute to the reduction of the soil $CH_4$ sink. However, N deposition in Europe and the United States has shifted from an increase to a decrease since early or middle 1990s (Du et al., 2014b; Waldner et al., 2014), implying a possible bounce back of soil $CH_4$ sink since then. Moreover, our results indicate that the response ratio of growing-season soil $CH_4$ uptake to N deposition is significantly regulated by growing-season mean temperature ($p$=0.017), implying an interaction between climate warming and N deposition to determine soil $CH_4$ sink. Although soil moisture is an important

regulator of soil $CH_4$ flux (Le Mer and Roger, 2001; Kaiser et al., 2018), we found no significant regulation of growing-season monthly mean precipitation on the response ratio of growing-season soil $CH_4$ flux to N deposition ($p$=0.36), which suggests no constraint of soil moisture during growing season across these forest biomes. Better understanding of the mechanisms behind the interaction of N deposition and other global changing factors calls for more efforts on multifactorial experiments (Kardol et al., 2012). This will also help to improve the prediction of future change in global soil $CH_4$ sink.

Our results also have important implications for improving the performance of process-based models to simulate and quantify the effect of global N deposition on soil $CH_4$ sinks (e.g., Ridgwell et al., 1999; Curry, 2007; Murguia-Flores et al., 2018). Currently, the existing models simply account for a negative effect of N inputs on soil $CH_4$ uptake by involving an inhibition factor (Ridgwell et al., 1999; Curry et al., 2007; Murguia-Flores et al., 2018). Using a process-based model, Murguia-Flores et al. (2018) estimated that N inputs via atmospheric deposition and fertilization reduced global soil uptake

of atmospheric $CH_4$ by 1.38 Tg yr$^{-1}$. However, our analysis indicates that a positive effect of low-level N addition on soil $CH_4$ uptake can occur in boreal forest, implying that existing models may overestimate the negative effect of N deposition on




global soil $CH_4$ sinks. This indicates the necessity of models to separately consider the effect of low-level N addition from that of high-level N addition and distinguish the effects across forest biomes.

## 4. Conclusion

Our synthesis of experimental results indicates that growing-season soil $CH_4$ flux was significantly affected by N additions in global forest biomes. Soil $CH_4$ flux showed different response ratios to low-level N addition and high-level N addition. Specifically, low-level N addition significantly increased growing-season soil $CH_4$ sink in boreal forest, while it significantly decreased growing-season soil $CH_4$ sink in temperate and subtropical forests. Compared with previous assessments based on a meta-analysis approach (Liu and Greaver, 2009; Aronson and Helliker, 2010), this work improves our understanding of biome-specific effect of N deposition on soil $CH_4$ sink in global forests. Our estimate indicates a significant reduction of soil $CH_4$ sink by N deposition, although it is negligible compared with the total growing-season soil $CH_4$ sink in global forest. The effect of N deposition on soil $CH_4$ flux is poorly understood in tropical forest due to a lack of experimental studies. This leads to a major uncertainty in the overall effect of N deposition on soil $CH_4$ sink in global forest. To better understand future trend of soil $CH_4$ sink in global forest, more experimental and modelling efforts are needed to incorporate the effects of N deposition and other global changing factors, especially in tropical forests.

## Author contribution

E.D. conceived the idea. E.D. and N.X. compiled the database and analyzed the data. E.D., N.X. and W.D. wrote and revised the manuscript.

## Competing interests

The authors declare that they have no conflict of interest.

## Acknowledgement

This work was supported by Fok Ying-Tong Education Foundation (Grant No. 161015), National Natural Science Foundation of China (Grant No. 41877328) and Project of State Key Laboratory of Earth Surface and Resource Ecology of Beijing Normal University (Grant No. 2017-ZY-07).

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

| Forest biome | Area (Million ha) | $Flux_{GS}$ (mg $CH_4$ $m^{-2}$ $h^{-1}$) | $BioSink_{GS}$ (Tg $CH_4$ per GS) | Biome mean N deposition (kg N $ha^{-1}$ $yr^{-1}$) | $R_{CH4-N}$ g $CH_4$ $kg^{-1}$ N | $BioEff_{Ndep}$ (Tg $CH_4$ per GS) |
|---|---|---|---|---|---|---|
| Boreal | 1225 | -0.281±0.120[b] | -10.13±4.33 | 1.2 | -59.3±28.6[b] | -0.029±0.016 |
| Temperate | 673 | -0.050±0.016[a] | -1.74±0.56 | 7.3 | 8.9±5.5[a] | 0.025±0.015 |
| Subtropical | 320 | -0.023±0.006[a] | -0.64±0.17 | 14.6 | 10.8±3.1[a] | 0.051±0.014 |

10 Note: The area of each biome (base year 2010) was derived from FAO Global Forest Resources Assessment (Keenan et al., 2015). Mean N deposition (base year 2010) of

11 each forest biome was derived from Schwede et al. (2018). Same lowercase letters mean no significant difference of mean growing-season soil $CH_4$ flux between forest

12 biomes, and different letters mean significant difference.



13   **Table 2**. Effects of growing-season mean temperature (GSMT), growing-season mean monthly

14   precipitation (GSMP), nitrogen additions and their interactions on the response ratio of growing-season

15   soil $CH_4$ uptake to N additions.

| | Estimate | SE | t | p |
|---|---|---|---|---|
| (Intercept) | -397 | 178 | -2.23 | 0.031 |
| GSMT | 23.2 | 9.31 | 2.49 | *0.017* |
| GSMP | 1.29 | 1.40 | 0.92 | 0.363 |
| N addition | 2.89 | 0.82 | 3.51 | *0.001* |
| GSMT×GSMP | -0.084 | 0.068 | -1.23 | 0.225 |
| GSMT×N addition | -0.196 | 0.063 | -3.08 | *0.004* |
| GSMP×N addition | 0.007 | 0.004 | 1.51 | 0.139 |



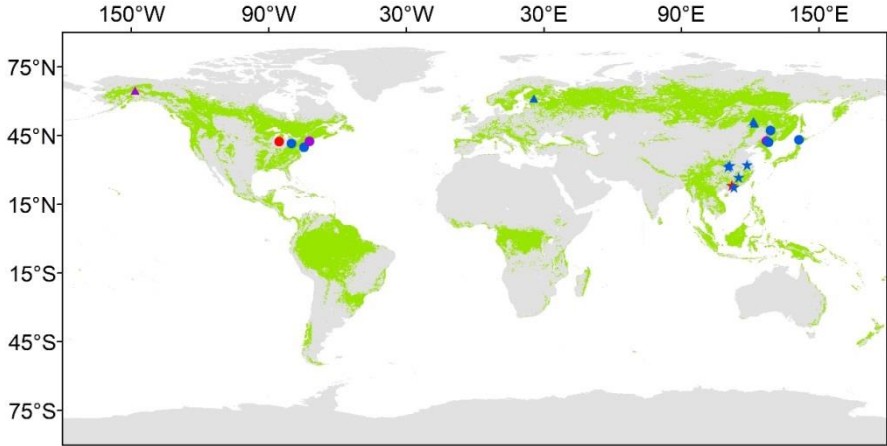

**Figure 1.** Geographical distribution of 25 forests receiving experimental nitrogen additions. The green shadows indicate the distribution of forest. The symbols triangle, circle and star indicate boreal, temperate and subtropical forests, respectively. The colors blue, purple and red indicate one, two and three experimental forests at one site, respectively.





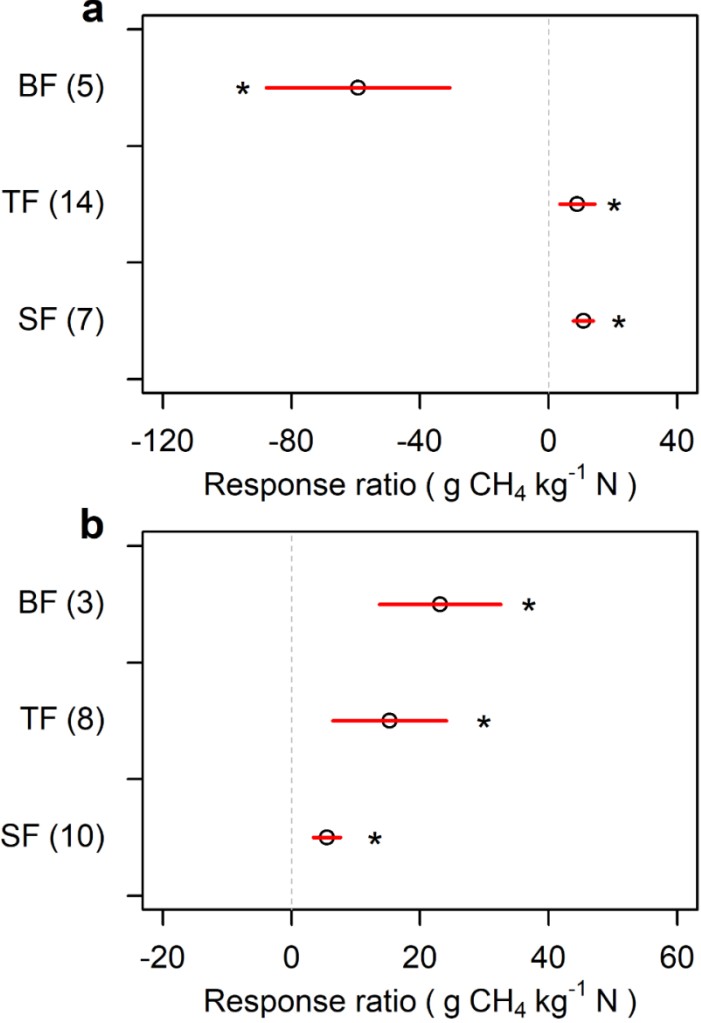

**Figure 2.** The response ratio of growing-season soil $CH_4$ flux to a) low-level N addition ($<60$ kg $N^{-1}$ $yr^{-1}$) and b) high-level N addition ($> 60$ kg $N^{-1}$ $yr^{-1}$) in boreal (BF), temperate (TF), and subtropical forests (SF), respectively. The asterisk (*) indicates a significant effect ($p<0.05$).