# Peer review of "Effects of nitrogen deposition on growing-season soil methane sink across global forest biomes"

_Biogeosciences, 2019_

## Referee Comment (RC1) · Anonymous Referee #1 · 3 Mar 2019

Dear authors,

I had the pleasure of reading your article entitled "Effects of nitrogen deposition on growing-season soil methane sink". The manuscript is definitely a good effort to investigate the effect of nitrogen deposition on atmospheric CH4 uptake by soils across forested ecosystems. This topic is very hot in the literature at the moment. The manuscript is very well written and easy to read and has a short but sufficient extent. That said, I found some fundamental flaws in three different aspects of the manuscripts: the assumptions you made by extrapolating your results, the datasets you employ and how the data was collected, and some problems with the general structure.

Major comments

[Figure]

• The definition and usage of the growing season extension is not well justified, nor correctly employed. Firstly, you never state why you are using the CH4 uptake of the growing season and not for the whole year. Most of the papers you revise have data for more than a year. Also, the growing season varies greatly across and within ecosystems, thus cannot be set to a single value by biome. Growing season can be defined by multiple variables (temperature, precipitation, number of frozen days), thus the usage of a simple single value for each biome is simply not acceptable. As a result, there is no justification to consider the effect of nitrogen over the CH4 uptake only during the growing season. Why not simply consider the whole year? If the growing season has importance for your analysis it is not reflected correctly in the manuscript and is never justified.

• Secondly and possible the strongest criticism, your assumptions about the positive effect of nitrogen in the soil CH4 sink in the boreal forest cannot be sustained with just four papers. The results are a wild extrapolation from studies that are not sufficient, nor analyzed correctly in your literature review. To be more precise: the work of Gulledge and Schimel (2000) found that nitrogen inhibit CH4 consumption in boreal forest; Maljanen et al. (2006) found perform a factorial experiment with nitrogen and ashes and while nitrogen alone did increase the uptake it was not statistically different; additionally, ashes with nitrogen decrease the CH4 uptake. Xu et al., 2014, found and increase in the uptake in the lowest N concentration (10 kg N ha$-1$ year$-1$) but a negative effect with 20 and 40 kg N ha$-1$ year$-1$ which are in the low N category you consider and do not be reflected in the results you obtained (why is the BF bar in figure 2 not going all the way to negative number based on this?). Finally, your fourth work Gao et al., 2013 is not available on google scholar. Based on this, the evidence to argue that boreal forest (the only ecosystem) presented an increased uptake due to nitrogen is not sustained at all. The other two ecosystems presented a decrease in all conditions, which is not novel.

• Thirdly, your definition of the of low and high N categories seems completely arbi-

trary and not justified based on literature. Why this threshold and not another?

• Finally, you are extrapolating data from 5-10 points, which are highly aggregated spatially to assume a biome-level behavior, which is incorrect. In other literature reviews of the topic, the authors revise around 35 papers to propose general mechanisms that can be scaled. In other words, you are trying to extrapolate a pattern based on few observations, without the proposition of an underlying mechanisms to support the increase in spatial scale.

Minor comments

• There are some mistakes defining the sign for the CH4 sink (both negative and positive signs are used along the paper). It needs to be consistent.

• Page 2, line 27, you started to talk about China as a hot spot for N deposition with no previous justification and not using this particular region in the paper. If you are focusing on a global scale, you should either give more examples or eliminate the regional-level comparisons.

• Page 6, line 25, the argument about the CO2 equivalents to measure the N effect over soil CH4 sink is absolutely out of place. 1) You cannot predict the effect of N deposition in the next 100 years using current values, 2) you are using a GHG potential of 25 (should be 28), 3) you create very strong arguments with very little evidence and not sufficient data, 4) finally, CO2-eq are not really used any longer, as the relationship of GHG to CO2 is not linear.

---

## Short Comment (SC1) · 13 Mar 2019

Dear authors,

it was a pleasure to read your very interesting discussion article.

I would like to make a comment. You wrote in Sect 2.1. "We excluded experiments by applying urea and manure, because these organic N forms have limited implications for the effects of N deposition (Aronson and Helliker, 2010) as N deposition mainly occurs in forms of inorganic N (Vet et al., 2014)." Based on this statement, you excluded one of the most long-term measurements of the effects of N addition on soil CH4 fluxes from tropical forests (4 years of data, Veldkamp et al., 2013) (as stated on P7/L5 of the discussion paper).

1) Could you please clarify more specifically where in the cited paper (Aronson and Helliker, 2010) you refer to when arguing that urea has limited implications for the effects of N deposition. When reading that paper (Aronson and Helliker, 2010), I find discussion on the form of N added on P3249-3250. The authors state there that "The effects of N addition could be similar regardless of N form used, due to the presence of microorganisms capable of rapid N transformation by variation in microbial consortia. The timing of fertilization may determine the form of N that methane-cycling microorganisms encounter in the soil more than the actual N species added. The fact that the addition of urea and ammonium nitrate were capable of increasing nitrate availability significantly in Delgado et al. (1996) underscores this point. Therefore, any conclusions of the effects of the specific N species relative to others must be highly qualified, as the form of N that results may be quite different from that added."

In terms of the tropical N-addition study that was excluded from your review, there is detailed data on soil extractable nitrate and ammonium concentrations that shows that urea additions have chronically changed the soil inorganic N status of the studied forests (Koehler et al., 2009). Therefore, I would like to challenge your decision to exclude the study on soil CH4 fluxes from the same sites (Veldkamp et al., 2013), based on the argument that N was added in the form of urea.

2) In that same study that you excluded from your review (Veldkamp et al., 2013), N addition did not decrease soil CH4 uptake, which is in conflict with the conclusion you made in your discussion article (e.g. abstract "However, high-level N addition significantly decreased growing-season soil CH4 uptake across boreal, temperate, and subtropical forests."). N-addition to the tropical forest sites in Panama that you excluded from your review (Veldkamp et al., 2013) did not affect soil CH4 fluxes.

I suggest that you include that detailed study on the effects of N-addition on CH4 fluxes from tropical forest soils, rather than excluding it with the argument that "the results based on N additions by using urea shed limited lights on the effect of N deposition" (P7/L5-6). Further information on detailed and long-term soil CH4 profiles (Koehler

et al., 2012) may assist in discussing that study in the context of your review, and in discussing why the N-addition effects on CH4 fluxes in these sites may differ from the overall results you found in your review.

Yours sincerely,

Birgit Koehler

References Aronson E.L. and Helliker B.R.: Methane flux in non-wetland soils in response to nitrogen addition: a meta-analysis. Ecology 91(11), 3242-3251, 2010.

Koehler B., Corre M.D., Veldkamp E., Wullaert H., Wright S.J.: Immediate and long-term nitrogen oxide emissions from tropical forest soils exposed to elevated nitrogen input. Global Change Biology 15(8), 2049-2066, 2009

Koehler B., Corre M.D., Steger K., Well R., Zehe E., Sueta J.P., Veldkamp E.: An in-depth look into a tropical lowland forest soil: nitrogen-addition effects on the contents of N2O, CO2 and CH4 and N2O isotopic signatures down to 2-m depth. Biogeochemistry 111, 695-713, 2012; Erratum Biogeochemistry 111, 715-717, doi: 10.1007/s10533-012-9780-6, 2012

Veldkamp E., Koehler B., Corre M.D.: Indications of nitrogen-limited methane uptake in tropical forest soils. Biogeosciences 10, 5367-5379, 2013

---

## Referee Comment (RC2) · Anonymous Referee #2 · 19 Mar 2019

This manuscript reports on a synthesis or meta-study of studies that report the effects of N additions on soil CH4 fluxes. The authors did a literature search using several sources, calculated a response variable, stratified results according to dose and biome, and conclude that low-level N addition increased Ch4 uptake in boreal forest while decreases occurred at high N addition levels and in all other biomes.

A meta-study is only as good as the sources where the information comes from, and therefore studies used in such a meta-study should be generally accessible and peer-reviewed. Unfortunately, several of the studies used in this meta-study did not meet these criteria. I find it problematic that apart from ISI Web of Science, also Google Scholar and China National knowledge Infrastructure are used to search for literature. To my knowledge, these two latter sources also list reports that are not peer-reviewed.

[Figure]

I strongly suggest to ONLY use ISI as a literature source, because here only peer-reviewed sources are listed. Studies that are not peer reviewed should not be included in a synthesis. In your meta-study I suggest to exclude the following studies listed in the supplement: - two of the studies listed are a MSc thesis (Wang, 2012, Pan, 2013). Such a thesis is not considered peer-reviewed, please exclude them. -all studies that involved incubations, instead of field measurements. For example, the study by Chen et al. 2017 mentions that they did laboratory incubations. You even state in Page 3, line 16 that you only included studies that used closed static chamber technology. Apparently that is not true. There may be more studies with incubations, I did not check them all. - Please, exclude studies published in Chinese (or other non-English publications) with only an English abstract. I do not consider such studies as generally accessible. For example, the study by Hu et al., 2011, is only accessible in Chinese and there may be more in the list.

You use the 60 kg N ha-1 yr-1 as an arbitrary cut-off between 'low' and 'high' level N addition. Did you calculate also the background N-deposition in the N additions? For example, the study by Li et al., 2015, mentions that there is a background N deposition of more than 30 kg N ha-1 yr-1, while the treatment is 40 kg ha-1 yr-1. Together this would be more than 60 kg N ha-1 yr-1 and the 40 kg treatment should be grouped as 'high' level N addition. I suspect that is not how you did this and it just illustrates how arbitrary the choice of 60 kg N ha-1 yr-1 is.

You did not mention any other criteria for inclusion or exclusion of studies. However, I think you should define what you consider a sufficiently large plot and a sufficiently wide buffer zone between treatments. Also, were all studies having true replicates? This kind of important information on the quality of studies is completely ignored in your manuscript.

I found it very adventurous that you excluded the only peer-reviewed study conducted in tropical ecosystems (Veldkamp et al., 2013) with the argument that urea as an organic N form has 'limited implications for the effects of N deposition', then cite Aronson and

Helliker, (2010) as the source for this statement (page 3, line 20), and later lament that there are no studies conducted in tropical areas and then even extrapolate the results from subtropical forests to the tropics. -First of all, while urea is strictly spoken an organic N source, it is quickly hydrolysed ($NH_2CONH_2 + H_2O \rightarrow CO_2 + 2NH_3$) after application and the gaseous $NH_3$ reacts with water to form ammonium ($NH_4+$). Only on soils with a high pH there are significant losses through volatilization. -Second, in the paper by Aronson & Helliker (2010), I did not find any statement that urea has limited implications for the effects of N deposition. In contrast, they also analysed studies with urea additions. They found no difference between Urea and other pure N fertilizers. They also concluded that 'any conclusions of the effects of specific N species relative to others must be highly qualified, as the form of N that results may be quite different from that added'. Therefore, to quote the Aronson & Helliker (2010) paper as the source why studies that add urea should be excluded is misleading. -Third, ignoring the only tropical study and later filling up the gap with studies from subtropical areas is very adventurous. Especially since the study conducted in the tropics did not following the hypothesized trend across biomes.

The hypothesis to be tested (Page 2, line 21 and further) is weak and based on incomplete assumptions. You simply assume that N availability increases from boreal to tropical biomes, which is not true. If you read publications by Vitousek more carefully you will see that the main factor is not the biome but how heavily weathered soils are. More than half of the tropics is located on soils that are not heavily weathered (e.g. montane forests) and N availability is expected to be as low as other biomes where young, less weathered soils dominate.

You group all forest ecosystems together and make no difference between natural forerst and plantations or managed forests. Tree plantations typically have significant growth rates and are almost always N limited, also in tropical and subtropical conditions. Ignoring this may lead to wrong conclusions.

In summary, this synthesis is poorly conducted. There are studies included that are not

peer-reviewed and there were no quality criteria for the studies that were included. The hypothesis is weak and based on incomplete assumptions. The distinction between 'high level' and 'low level' N addition is arbitrary and the background N deposition is ignored. Finally, I could not find any objective reason why studies where urea was added were excluded. Given these weaknesses of this synthesis I strongly doubt the validity of the conclusions and I recommend not to publish this manuscript in Biogeosciences.

---

## Author Comment (AC1) · 17 Apr 2019

**Reply to the comments by Birgit Koehler**

Enzai Du[1,2*], Nan Xia[2], Wim de Vries[3,4]

[1]State Key Laboratory of Earth Surface Processes and Resource Ecology, Faculty of Geographical Science, Beijing Normal University, Beijing100875, China

[2]School of Natural Resources, Faculty of Geographical Science, Beijing Normal University, Beijing100875, China

[3]Wageningen University and Research, Environmental Research, PO Box 47, NL-6700 AA Wageningen, the Netherlands

[4]Wageningen University and Research, Environmental Systems Analysis Group, PO Box 47, NL-6700 AA Wageningen, the Netherlands

*Correspondence to*: Enzai Du (enzaidu@bnu.edu.cn)

**Comment**: It was a pleasure to read your very interesting discussion article. I would like to make a comment. You wrote in Sect 2.1. "We excluded experiments by applying urea and manure, because these organic N forms have limited implications for the effects of N deposition (Aronson and Helliker, 2010) as N deposition mainly occurs in forms of inorganic N (Vet et al., 2014)." Based on this statement, you excluded one of the most long-term measurements of the effects of N addition on soil CH4 fluxes from tropical forests (4 years of data, Veldkamp et al., 2013) (as stated on P7/L5 of the discussion paper).

**Reply**: Thanks for your helpful comments. According to your suggestion, we have included results of urea addition experiments in the revised manuscript. We have also conducted a meta-analysis to test the significance of nitrogen addition effect on soil $CH_4$ flux. Please see more information in our detailed reply to your specific comments.

**Comment**: 1) Could you please clarify more specifically where in the cited paper (Aronson and Helliker, 2010) you refer to when arguing that urea has limited implications for the effects of N deposition. When reading that paper (Aronson and Helliker, 2010), I find discussion on the form of N added on P3249-3250. The authors state there that "The effects of N addition could be similar regardless of N form used, due to the presence of microorganisms capable of rapid N transformation by variation in microbial consortia. The timing of fertilization may determine the form of N that methane-cycling microorganisms encounter in the soil more than the actual N species added. The fact that the addition of urea and ammonium nitrate were capable of increasing nitrate availability significantly in Delgado et al. (1996) underscores this point. Therefore, any conclusions of the effects of the specific N species relative to others must be highly qualified, as the form of N that results may be quite different from that added."

**Reply**: In the paper by Aronson and Helliker (2010), the authors showed that the effects of N fertilization on soil $CH_4$ uptake did vary significantly with N forms (Figure R1). Specifically, the effects of urea and ammonium nitrate showed slight overlap, although the statistical analysis indicated no significant difference (Figure R1). This result motivated us to exclude experiments with urea additions in our previous manuscript, because a) N deposition mainly occurs in inorganic N forms, and b) urea might be less capable to indicate the effects of N deposition. As suggested by you and other reviewers, we have included urea addition experiments in the revised manuscript in view of the fact that the effects of urea and ammonium nitrate were not significantly different.

[Figure]

Figure R1. Effects of different N forms on soil $CH_4$ uptake (Aronson and Helliker, 2010)

**Reference**:

Aronson, E.L., Helliker, B.R. 2010. Methane flux in non-wetland soils in response to nitrogen addition: a meta-analysis. Ecology, 91(11), 3242-3251.

**Comment**: In terms of the tropical N-addition study that was excluded from your review, there is detailed data on soil extractable nitrate and ammonium concentrations that shows that urea additions have chronically changed the soil inorganic N status of the studied forests (Koehler et al., 2009). Therefore, I would like to challenge your decision to exclude the study on soil $CH_4$ fluxes from the same sites (Veldkamp et al., 2013), based on the argument that N was added in the form of urea.

**Reply:** We generally agree to include the studies with urea additions as the effects of urea and ammonium nitrate were not significantly different (Aronson and Helliker, 2010). In the revised manuscript, we have updated our database by including reported results on urea addition experiments. Overall, 6 urea based experiments are included, including one from temperate forest (Geng et al.,2017), one from subtropical forest (Zhang et al.,2017), and four from tropical forest (Matson et al.,2016; Veldkamp et al.,2013; Mori et al. 2017) (see more detailed references as below).

**References**:

Aronson, E.L., Helliker, B.R. 2010. Methane flux in non‐wetland soils in response to nitrogen addition: a meta-analysis. Ecology, 91(11), 3242-3251.

Geng, J., Cheng, S., Fang, H., Yu, G., Li, X., Si, G., et al. (2017). Soil nitrate accumulation explains the nonlinear responses of soil $CO_2$ and $CH_4$ fluxes to nitrogen addition in a temperate needle-broadleaved mixed forest. Ecological Indicators, 79, 28-36.

Matson, A. L. , Corre, M. D. , & Veldkamp, E. . (2016). Canopy soil greenhouse gas dynamics in response to indirect fertilization across an elevation gradient of tropical montane forests. Biotropica, 49.

Mori, T., Imai, N., Yokoyama, D., Mukai, M., & Kitayama, K. (2017). Effects of selective logging and application of phosphorus and nitrogen on fluxes of $CO_2$, $CH_4$ and $N_2O$ in lowland tropical rainforests of Borneo. Journal of Tropical Forest Science, 248-256.

Veldkamp, E., Koehler, B., & Corre, M. D. (2013). Indications of nitrogen-limited methane uptake in tropical forest soils. Biogeosciences, 10(8), 5367-5379.

Zhang, K., Zheng, H., Chen, F., Li, R., Yang, M., Ouyang, Z., et al. (2017). Impact of nitrogen fertilization on soil–Atmosphere greenhouse gas exchanges in eucalypt plantations with different soil characteristics in southern China. PloS One, 12(2), e0172142.

**Comment**: 2) In that same study that you excluded from your review (Veldkamp et al., 2013), N addition did not decrease soil $CH_4$ uptake, which is in conflict with the conclusion you made in your discussion article (e.g. abstract "However, high-level N addition significantly decreased growing-season soil $CH_4$ uptake across boreal, temperate, and subtropical forests."). N-addition to the tropical forest sites in Panama that you excluded from your review (Veldkamp et al., 2013) did not affect soil $CH_4$ fluxes.

**Reply:** Thanks for your comments. We have realized that speculating the effects of N additions in tropical forests based on results from subtropical forests is not appropriate. In the revised manuscript, we have updated our database and tested the significance of nitrogen addition effect on soil $CH_4$ flux by conducting a meta-analysis in R software with metaphor package (Viechtbauer, 2010). We used a mean difference ($Flux_{treatment}$-$Flux_{control}$, the difference of mean growing-season soil $CH_4$ fluxes between the treatment plots and control plots) as the effect size to evaluate the effect of N additions. Based on 4 experiments from tropical forest (Veldkamp et al., 2013; Matson et al., 2016; Mori et al., 2017), we show that simulated N deposition ($<60$ kg $N^{-1}$ $yr^{-1}$) and high-level N addition ($<60$ kg $N^{-1}$ $yr^{-1}$) both had no significant effect on soil $CH_4$ uptake (Figure R2a&b). Overall, no

significant effect of N additions on soil $CH_4$ uptake was found in tropical forest (Figure R2c). We have revised the manuscript accordingly.

[Figure]

5    Figure R2. The mean difference (and 95% confidence intervals) of growing-season soil $CH_4$ flux to a) simulated N deposition (<60 kg $N^{-1}$ $yr^{-1}$), b) high-level N addition (> 60 kg $N^{-1}$ $yr^{-1}$) and c) all N treatments in boreal (BF), temperate (TemF), subtropical (STF), and tropical forest (TroF), respectively. The asterisk (*) indicates a significant effect (p<0.05).

**Reference**

Matson, A. L. , Corre, M. D., Veldkamp, E. 2016. Canopy soil greenhouse gas dynamics in response to indirect fertilization across an elevation gradient of tropical montane forests. Biotropica, 49(2), 153-159.

Mori, T., Imai, N., Yokoyama, D., Mukai, M., Kitayama, K. 2017. Effects of selective logging and application of phosphorus and nitrogen on fluxes of $CO_2$, $CH_4$ and $N_2O$ in lowland tropical rainforests of borneo. Journal of Tropical Forest Science, 248-256.

Veldkamp, E., Koehler, B., Corre, M. D. 2013. Indications of nitrogen-limited methane uptake in tropical forest soils. Biogeosciences, 10(8), 5367-5379.

Viechtbauer, W. 2010. Conducting meta-analyses in R with the metafor package. Journal of Statistical Software, 36(3), 1-48.

**Comment**: I suggest that you include that detailed study on the effects of N-addition on $CH_4$ fluxes from tropical forest soils, rather than excluding it with the argument that "the results based on N additions by using urea shed limited lights on the effect of N deposition" (P7/L5-6). Further information on detailed and long-term soil $CH_4$ profiles (Koehler et al., 2012) may assist in discussing that study in the context of your review, and in discussing why the N-addition effects on $CH_4$ fluxes in these sites may differ from the overall results you found in your review.

**Reply:** Thanks for your suggestion. As in our reply above, we have conducted a reanalysis by including experiments with urea additions in tropical forest. The current results indicate that N additions have no significant effects on soil $CH_4$ fluxes (Fig. R2). As N deposition in many tropical regions is expected to increase in the future (Lamarque et al., 2013), further efforts are needed to evaluate the effect of N deposition on soil $CH_4$ flux in tropical forest by conducting extra N addition experiments with low-level additions of inorganic N. We have also revised the result section and cited Koehler et al. (2009 &2012) when discussing the potential mechanisms in tropical forest.

**Reference**

Lamarque, J.F., Dentener, F., McConnell, J., Ro, C.-U., Shaw, M., Vet, R., Bergmann, D., Cameron-Smith, P., Dalsoren, S., Doherty, R., Faluvegi, G., Ghan, S. J., Josse, B., Lee, Y. H., MacKenzie, I. A., Plummer, D., Shindell, D. T., Skeie, R. B., Stevenson, D. S., Strode, S., Zeng, G., Curran, M., Dahl-Jensen, D., Das, S., Fritzsche, D., and Nolan, M. 2013. Multi-model mean nitrogen and sulfur deposition from the Atmospheric Chemistry and Climate Model Intercomparison Project (ACCMIP): evaluation of historical and projected future changes, Atmos. Chem. Phys., 13, 7997-8018.

Yours sincerely,

Birgit Koehler

References

Aronson E.L. and Helliker B.R.: Methane flux in non-wetland soils in response to nitrogen addition: a meta-analysis. Ecology 91(11), 3242-3251, 2010.

Koehler B., Corre M.D., Veldkamp E., Wullaert H., Wright S.J.: Immediate and long-term nitrogen oxide emissions from tropical forest soils exposed to elevated nitrogen input. Global Change Biology 15(8), 2049-2066, 2009

Koehler B., Corre M.D., Steger K., Well R., Zehe E., Sueta J.P., Veldkamp E.: An indepth look into a tropical lowland forest soil: nitrogen-addition effects on the contents of $N_2O$, $CO_2$ and $CH_4$ and $N_2O$ isotopic signatures down to 2-m depth. Biogeochemistry 111, 695-713, 2012; Erratum Biogeochemistry 111, 715-717, doi: 10.1007/s10533- 012-9780-6, 2012

Veldkamp E., Koehler B., Corre M.D.: Indications of nitrogen-limited methane uptake in tropical forest soils. Biogeosciences 10, 5367-5379, 2013

10

---

## Author Comment (AC2) · 17 Apr 2019

**Reply to the comments by reviewer #1**

Enzai Du[1,2*], Nan Xia[2], Wim de Vries[3,4]

[1]State Key Laboratory of Earth Surface Processes and Resource Ecology, Faculty of Geographical Science, Beijing Normal University, Beijing100875, China

[2]School of Natural Resources, Faculty of Geographical Science, Beijing Normal University, Beijing100875, China

[3]Wageningen University and Research, Environmental Research, PO Box 47, NL-6700 AA Wageningen, the Netherlands

[4]Wageningen University and Research, Environmental Systems Analysis Group, PO Box 47, NL-6700 AA Wageningen, the Netherlands

*Correspondence to*: Enzai Du (enzaidu@bnu.edu.cn)

**Comment**:I had the pleasure of reading your article entitled "Effects of nitrogen deposition on growing-season soil methane sink". The manuscript is definitively a good effort to investigate the effect of nitrogen deposition on atmospheric $CH_4$ uptake by soils across forested ecosystems. This topic is very hot in the literature at the moment. The manuscript is very well written and easy to read and has a short but sufficient extent. That said, I found some fundamental flaws in three different aspects of the manuscripts: the assumptions you made by extrapolating your results, the datasets you employ and how the data was collected, and some problems with the general structure.

**Reply**: Thanks for your helpful comments. We have substantially improved the manuscript according to your suggestions. Please see more detailed information in our reply to your specific comments. We believe that the main concerns have been well addressed in the revised manuscript.

**Comment**: The definition and usage of the growing season extension is not well justified, nor correctly employed. Firstly, you never state why you are using the $CH_4$ uptake of the growing season and not for the whole year. Most of the papers you revise have data for more than a year. Also, the growing season varies greatly across and within ecosystems, thus cannot be set to a single value by biome. Growing season can be defined by multiple variables (temperature, precipitation, number of frozen days), thus the usage of a simple single value for each biome is simply not acceptable. As a result, there is no justification to consider the effect of nitrogen over the $CH_4$ uptake only during the growing season. Why not simply consider the whole year? If the growing season has importance for your analysis it is not reflected correctly in the manuscript and is never justified.

**Reply:** Our analysis focuses on the effects of N additions on growing-season soil $CH_4$ uptake because a) growing-season $CH_4$ uptake accounts for a major proportion of the annual $CH_4$ sink (Le Mer and Roger, 2001) and b) the reported data in literature are usually measured during the growing season. As required by other reviewers, we have updated our database by a) excluding reports from theses without full peer-review, and b)

including results of field experiments using urea additions. The updated database includes experimental results of 28 forests across 22 sites (Fig.R1), but only 14 of the 28 experiments measured whole-year soil $CH_4$ uptake (0/7 for boreal forest, 0/7 for temperate forest, 10/10 for subtropical forest, 4/4 for tropical forests). Although some papers reported data for more than one year, they only measured soil $CH_4$ flux during the growing season of each year. When basing on measurements of annual soil $CH_4$ uptake, we are not able to conduct a statistical analysis for boreal and temperate forests. Due to the reasons discussed above, our analysis assesses the effects of N deposition on growing-season soil $CH_4$ uptake.

In our manuscript, mean growing season length was used to estimate the total growing-season soil $CH_4$ sinks and their response to N deposition for each biome. Generally, growing season can be defined by several approaches, including a) thresholds of multiple climatic variables, b) field monitoring of plant phenology and c) remote sensing of plant phenology. The estimated growing season length usually varies with different approaches. Moreover, we also understand that there are variations of growing season extension in a forest biome. Based on an assessment in northern hemisphere (Piao et al., 2007), growing season extension generally varies from 3 to 5 months in boreal forest and 6 to 8 months in temperate forest (compare Fig. R1 with Fig. R2). We thus used mean growing season length of 4 (mid May to mid Sep.) and 7 (Apr. to Oct.) months for boreal and temperate forest, respectively. In the revised manuscript, we have now also discussed the uncertainties of growing season length due to different approaches and the inner biome variations. Thanks for your understanding.

[Figure]

Figure R1. Geographical distribution of 22 forested sites with 28 forests receiving experimental nitrogen additions. Green shadows indicate the distribution of global forest.

[Figure]

Figure R2. Growing season length of northern hemisphere (Piao et al., 2007)

**Reference**

Le Mer, J., Roger, P. 2001. Production, oxidation, emission and consumption of methane by soils: a review. European Journal of Soil Biology, 37(1), 25-50.

Piao, S., Friedlingstein, P., Ciais, P., Viovy, N., Demarty, J. 2007. Growing season extension and its impact on terrestrial carbon cycle in the Northern Hemisphere over the past 2 decades. Global Biogeochemical Cycles, 21(3): GB3018.

**Comment**: Secondly and possible the strongest criticism, your assumptions about the positive effect of nitrogen in the soil $CH_4$ sink in the boreal forest cannot be sustained with just four papers. The results are a wild extrapolation from studies that are not sufficient, nor analyzed correctly in your literature review. To be more precise: the work of Gulledge and Schimel (2000) found that nitrogen inhibit $CH_4$ consumption in boreal forest; Maljanen et al. (2006) found perform a factorial experiment with nitrogen and ashes and while nitrogen alone did increase the uptake it was not statistically different; additionally, ashes with nitrogen decrease the $CH_4$ uptake. Xu et al., 2014, found and increase in the uptake in the lowest N concentration (10 kg N ha$^{-1}$ year$^{-1}$) but a negative effect with 20 and 40 kg N ha$^{-1}$ year$^{-1}$ which are in the low N category you consider and do not be

reflected in the results you obtained (why is the BF bar in figure 2 not going all the way to negative number based on this?). Finally, your fourth work Gao et al., 2013 is not available on google scholar. Based on this, the evidence to argue that boreal forest (the only ecosystem) presented an increased uptake due to nitrogen is not sustained at all. The other two ecosystems presented a decrease in all conditions, which is not novel.

5  **Reply:** Thanks for your helpful comments. We have realized that our previous analysis based on Student's t-test is not appropriate because it failed to consider the inter-study heterogeneity variance ($\tau^2$) and the within-study variances ($\delta^2$). This leads to inaccurate statistical speculations that are misleading, such as the results for boreal forest in our earlier manuscript. In the revised manuscript, we have tested the significance of N addition effect on soil $CH_4$ flux by conducting a meta-analysis in R software with the mixed effect model in metaphor package

10  (Viechtbauer, 2010). We used a mean difference ($Flux_{treatment}$-$Flux_{control}$, the difference of mean growing-season soil $CH_4$ fluxes between the treatment plots and control plots) as the effect size to evaluate the effect of N additions. The heterogeneity variance ($\tau^2$) across sites and the within-study variances ($\delta^2$) were properly considered. If N additions result in a significant effect, we then estimated the response ratio of growing-season soil $CH_4$ flux based on the mean difference and the mean levels of N treatment.

15  Our reanalysis indicates that simulated N deposition  ($<60$ kg N $ha^{-1}$ $yr^{-1}$) only results in a significant decrease of soil $CH_4$ uptake in temperate forests, while no significant effects are found in boreal, subtropical and tropical forests (Fig. R3a). When receiving high-level N additions ($>60$ kg N $ha^{-1}$ $yr^{-1}$), soil $CH_4$ uptake is significantly decreased in boreal, temperate and subtropical forests (Fig. R3b). If all N treatments are compiled, the overall effects are similar to those of high-level N additions. In summary, only temperate forest shows a significant

20  response ratio to simulated N deposition, while high-level N additions result in significant response ratios across boreal, temperate and subtropical forests (Fig. R4). In view of the fact that N deposition is generally low in global forest ecosystems and rarely exceeding a maximum of 60 kg N $ha^{-1}$ $yr^{-1}$(Vet et al., 2014; Schwede et al., 2018), our results imply that separating the effects of N deposition and high-level N fertilization is necessary to avoid overestimate the effects of global N deposition. However, this has never been considered by existing studies at a

25  global scale (Liu and Greaver, 2009; Aronson and Helliker, 2010).

[Figure]

Figure R3. The mean difference (and 95% confidence intervals) of growing-season soil $CH_4$ flux to a) simulated N deposition (<60 kg $N^{-1}$ $yr^{-1}$), b) high-level N addition (> 60 kg $N^{-1}$ $yr^{-1}$) and c) all N treatments in boreal (BF), temperate (TemF), subtropical (STF), and tropical forest (TroF), respectively. The asterisk (*) indicates a significant effect (p<0.05).

[Figure]

Figure R4. The response ratio of growing-season soil $CH_4$ flux to simulated N deposition (LN, <60 kg $N^{-1}$ $yr^{-1}$), high-level N addition (HN, > 60 kg $N^{-1}$ $yr^{-1}$) and all N treatments (ALL) in boreal (BF), temperate (TemF) and subtropical forest (STF), respectively. Response ratio of insignificant effect (N treatments in tropical forest and LN in boreal forest) is not shown.

**Reference**

Aronson, E.L., Helliker, B.R. 2010. Methane flux in non‑wetland soils in response to nitrogen addition: a meta-analysis. Ecology, 91(11), 3242-3251.

Liu, L.L., and Greaver, T.L. 2009. A review of nitrogen enrichment effects on three biogenic GHGs: the $CO_2$ sink may be largely offset by stimulated $N_2O$ and $CH_4$ emission. Ecology Letters, 12, 1103–1117.

Schwede, D.B., Simpson, D., Tan, J., Fu, J., Dentener, F., Du, E., and De Vries, W.2018. Spatial variation of modelled total, dry and wet nitrogen deposition to forests at global scale. Environmental Pollution, 243, 1287-1301.

Vet, R., Artz, R.S., Carou, S., Shaw, M., Ro, C,U., Aas, W. et al. 2014.A global assessment of precipitation chemistry and deposition of sulfur, nitrogen, sea salt, base cations, organic acids, acidity and pH, and phosphorus, Atmospheric Environment, 93, 3–100.

Viechtbauer, W. 2010. Conducting meta-analyses in R with the metafor package. Journal of Statistical Software, 36(3), 1-48.

**Comment**: Thirdly, your definition of the low and high N categories seems completely arbitrary and not justified based on literature. Why this threshold and not another?

**Reply:** We defined this threshold based on a global assessment of N deposition by the World Meteorological Organization (WMO) Global Atmosphere Watch programme (GAW) (Vet et al., 2014). This assessment shows a range of N deposition in various regions of the world, from 1~62.25 kg N ha$^{-1}$ yr$^{-1}$. Specifically, the maximum level of N deposition occurs in eastern and southern China (Vet et al., 2014). We thus use 60 kg N ha$^{-1}$ yr$^{-1}$ as a threshold for maximum N deposition, in order to distinguish it from N fertilization with extremely high levels of N additions.

**Reference**

Vet, R., Artz, R.S., Carou, S., Shaw, M., Ro, C,U., Aas, W. et al. 2014.A global assessment of precipitation chemistry and deposition of sulfur, nitrogen, sea salt, base cations, organic acids, acidity and pH, and phosphorus, Atmospheric Environment, 93, 3–100.

**Comment**: Finally, you are extrapolating data from 5-10 points, which are highly aggregated spatially to assume a biome-level behaviour, which is incorrect. In other literature reviews of the topic, the authors revise around 35 papers to propose general mechanisms that can be scaled. In other words, you are trying to extrapolate a pattern

based on few observations, without the proposition of an underlying mechanisms to support the increase in spatial scale.

**Reply**: We fully agree that extrapolating data from limited sites could introduce large uncertainties. In the revised manuscript, we focused more on the different effects of simulated N deposition and high-level N fertilization. By updating our database and conducting a meta-analysis, we show that the effects of simulated N deposition and high-level N fertilization can be significantly different across forest biomes. Based on current results, simulated N deposition only results in a reduction of soil $CH_4$ sink in temperate forest. This finding implies that existing meta-analyses at a global scale have likely overestimated the effect of N deposition (Liu and Greaver, 2009; Aronson and Helliker, 2010), in view of the fact that N deposition is generally low in global forest ecosystems and rarely exceeding a maximum of 60 kg N ha$^{-1}$ yr$^{-1}$ (Vet et al., 2014; Schwede et al., 2018). To demonstrate the magnitude of the overestimation, we have now estimated and discussed the possible overestimation by comparing the scaling results based on the response ratios of simulated N deposition and high-level N fertilization. We have also mentioned that this kind of scaling has large uncertainty, but our purpose here is to demonstrate that separating the effects of N deposition and high-level N fertilization is necessary to avoid overestimate the effects of global N deposition.

**Reference**

Aronson, E.L., Helliker, B.R. 2010. Methane flux in non‑wetland soils in response to nitrogen addition: a meta-analysis. Ecology, 91(11), 3242-3251.

Liu, L.L., and Greaver, T.L. 2009. A review of nitrogen enrichment effects on three biogenic GHGs: the $CO_2$ sink may be largely offset by stimulated $N_2O$ and $CH_4$ emission. Ecology Letters, 12, 1103–1117.

Schwede, D.B., Simpson, D., Tan, J., Fu, J., Dentener, F., Du, E., and De Vries, W.2018. Spatial variation of modelled total, dry and wet nitrogen deposition to forests at global scale. Environmental Pollution, 243, 1287-1301.

Vet, R., Artz, R.S., Carou, S., Shaw, M., Ro, C,U., Aas, W. et al. 2014.A global assessment of precipitation chemistry and deposition of sulfur, nitrogen, sea salt, base cations, organic acids, acidity and pH, and phosphorus, Atmospheric Environment, 93, 3–100.

**Minor comments**

**Comment**: There are some mistakes defining the sign for the $CH_4$ sink (both negative and positive signs are used along the paper). It needs to be consistent.

**Reply**: We have checked through the manuscript and used consistent signs for $CH_4$ sinks in the revised manuscript. Specifically, soil $CH_4$ sink/uptake is indicated by negative values. A positive value of the mean difference ($Flux_{treatment}$-$Flux_{control}$) and response ratio indicates a reduction of soil $CH_4$ sink/uptake. This has been clarified in the revised manuscript.

**Comment**: Page 2, line 27, you started to talk about China as a hot spot for N deposition with no previous justification and not using this particular region in the paper. If you are focusing on a global scale, you should either give more examples or eliminate the regional-level comparisons.

**Reply:** Based on a global assessment of N deposition by the World Meteorological Organization (WMO) Global

10    Atmosphere Watch programme (GAW) (Vet et al., 2014), N deposition generally shows a global range of $1\sim62.25$ kg N ha$^{-1}$ yr$^{-1}$. Specifically, the maximum level of N deposition occurs in eastern and southern China (Vet et al., 2014). We thus went into a bit more details of N deposition status in China. We have revised the manuscript accordingly to avoid any misunderstanding.

**Reference**

15    Vet, R., Artz, R.S., Carou, S., Shaw, M., Ro, C,U., Aas, W. et al. 2014.A global assessment of precipitation chemistry and deposition of sulfur, nitrogen, sea salt, base cations, organic acids, acidity and pH, and phosphorus, Atmospheric Environment, 93, 3–100.

**Comment**: Page 6, line 25, the argument about the $CO_2$ equivalents to measure the N effect over soil CH4 sink is

20    absolutely out of place. 1) You cannot predict the effect of N deposition in the next 100 years using current values, 2) you are using a GHG potential of 25 (should be 28), 3) you create very strong arguments with very little evidence and not sufficient data, 4) finally, $CO_2$-eq are not really used any longer, as the relationship of GHG to CO2 is not linear.

**Reply**: We generally agree with your points. In the revised manuscript, we have excluded the discussion based on

25    $CO_2$ equivalents for $CH_4$ sink. Thanks again for the helpful comments.

---

## Author Comment (AC3) · 17 Apr 2019

**Reply to the comments by reviewer #2**

Enzai Du[1,2*], Nan Xia[2], Wim de Vries[3,4]

[1]State Key Laboratory of Earth Surface Processes and Resource Ecology, Faculty of Geographical Science, Beijing Normal University, Beijing100875, China

5 [2]School of Natural Resources, Faculty of Geographical Science, Beijing Normal University, Beijing100875, China

[3]Wageningen University and Research, Environmental Research, PO Box 47, NL-6700 AA Wageningen, the Netherlands

[4]Wageningen University and Research, Environmental Systems Analysis Group, PO Box 47, NL-6700 AA Wageningen, the Netherlands

10 *Correspondence to*: Enzai Du ([enzaidu@bnu.edu.cn](mailto:enzaidu@bnu.edu.cn))

**Comments**: This manuscript reports on a synthesis or meta-study of studies that report the effects of N additions on soil CH4 fluxes. The authors did a literature search using several sources, calculated a response variable, stratified results according to dose and biome, and conclude that low-level N addition increased $CH_4$ uptake in boreal forest while decreases occurred at high N addition levels and in all other biomes. A meta-study is only as

15 good as the sources where the information comes from, and therefore studies used in such a meta-study should be generally accessible and peer-reviewed. Unfortunately, several of the studies used in this meta-study did not meet these criteria. I find it problematic that apart from ISI Web of Science, also Google Scholar and China National Knowledge Infrastructure are used to search for literature. To my knowledge, these two latter sources also list reports that are not peer-reviewed. I strongly suggest to ONLY use ISI as a literature source, because here only

20 peer-reviewed sources are listed. Studies that are not peer reviewed should not be included in a synthesis. In your meta-study I suggest to exclude the following studies listed in the supplement: - two of the studies listed are a MSc thesis (Wang, 2012, Pan, 2013). Such a thesis is not considered peer-reviewed, please exclude them. -all studies that involved incubations, instead of field measurements. For example, the study by Chen et al. 2017 mentions that they did laboratory incubations. You even state in Page 3, line 16 that you only included studies

25 that used closed static chamber technology. Apparently that is not true. There may be more studies with incubations, I did not check them all. - Please, exclude studies published in Chinese (or other non-English publications) with only an English abstract. I do not consider such studies as generally accessible. For example, the study by Hu et al., 2011, is only accessible in Chinese and there may be more in the list.

**Reply:** Thanks for your comments, which have helped us to improve the manuscript substantially. We fully agree

30 that the quality of data sources is very important for our meta-analysis. In the revised manuscript, we have updated the database by a) excluding experimental results from theses without fully peer-review (Wang, 2012; Pan, 2013), b) excluding experimental results from laboratory incubations (only Chen et al. 2017), c) excluding

an experimental study that didn't report values of variance/standardized error (Steudler et al., 1989), d) excluding results of experiments with plot area < 10m$^2$ or without buffer zones, and e) including more results of field experiments using urea additions. We further beg your understanding to include reports published in fully peer-reviewed Chinese journals because these publications are generally of good quality in science. Overall, we updated our database based on the criteria above and the current database includes results of 28 experiments across 22 sites (see Figure R1).

[Figure]

Figure R1. Geographical distribution of 28 forests receiving experimental nitrogen additions. Green shadows indicate the distribution of global forest.

**Reference**

Pan, D.R. 2013. Study on greenhouse gas emission for grassland soil below different forest soils under precipitation reduction and Nitrogen deposition in Shennongjia mountain. Gansu Agricultural University, Master thesis.

Steudler, P.A., Bowden, R.D., Melillo, J.M., Aber, J.D. 1989. Influence of nitrogen fertilization on methane uptake in temperate forest soils. Nature, 341(6240), 314-316.

Wang, R.N. 2012. Effects of simulated atmospheric nitrogen deposition on the exchange fluxes of greenhouse gases in the temperate forest soil. Beijing Forestry University, Master thesis

**Comments**: You use the 60 kg N ha$^{-1}$ yr$^{-1}$ as an arbitrary cut-off between 'low' and 'high' level N addition. Did you calculate also the background N-deposition in the N additions? For example, the study by Li et al., 2015, mentions that there is a background N deposition of more than 30 kg N ha$^{-1}$ yr$^{-1}$, while the treatment is 40 kg ha$^{-1}$

yr$^{-1}$. Together this would be more than 60 kg N ha$^{-1}$ yr$^{-1}$ and the 40 kg treatment should be grouped as 'high' level N addition. I suspect that is not how you did this and it just illustrates how arbitrary the choice of 60 kg N ha-1 yr-1 is.

**Reply:** Thanks for your comments. We defined the threshold of N deposition versus high-level N addition based on a global assessment of N deposition by the World Meteorological Organization (WMO) Global Atmosphere Watch programme (GAW) (Vet et al., 2014). This assessment shows a range of N deposition in various regions of the world, from 1~62.25 kg N ha$^{-1}$ yr$^{-1}$. Specifically, the maximum level of N deposition occurs in eastern and southern China (Vet et al., 2014). We thus used 60 kg N ha$^{-1}$ yr$^{-1}$ as a threshold for a possible maximum of N deposition, in order to distinguish it from N fertilization with extremely high levels of N additions. In the revised manuscript, we have conducted a meta-analysis (Viechtbauer, 2010) by using background N deposition as a moderator. We used a mean difference (Flux$_{treatment}$-Flux$_{control}$, the difference of mean growing-season soil CH$_4$ fluxes between the treatment plots and control plots) as the effect size to evaluate the effect of N additions. The results indicate that background N deposition has a significant influence on the effect of N additions (p=0.0006). However, a recent assessment of N deposition indicates that average N deposition varied significantly across global forest biomes, showing a trend subtropical forest (14.6 kg N ha$^{-1}$ yr$^{-1}$) > temperate forest (7.3 kg N ha$^{-1}$ yr$^{-1}$) and tropical forest (7.2 kg N ha$^{-1}$ yr$^{-1}$) > boreal forest (1.2 kg N ha$^{-1}$ yr$^{-1}$). Moreover, total N availability from mineralization, biological N fixation and N deposition generally shows a decrease from tropical forest to boreal forest (Cleveland et al., 2013). This hinders us to separating the effects of background N deposition/availability from the effect of forest biomes. We have revised the manuscript to avoid any misunderstanding.

Our reanalysis indicates that simulated N deposition (<60 kg N ha$^{-1}$ yr$^{-1}$) only results in a significant decrease of soil CH$_4$ uptake in temperate forests, while no significant effects are found in boreal, subtropical and tropical forests (Fig. R2a). When receiving high-level N additions (>60 kg N ha$^{-1}$ yr$^{-1}$), soil CH$_4$ uptake is significantly decreased in boreal, temperate and subtropical forests (Fig. R2b). If all N treatments are compiled, the overall effects are similar to those of high-level N additions. In summary, only temperate forest shows a significant response ratio to simulated N deposition, while high-level N additions result in significant response ratios across boreal, temperate and subtropical forests (Fig. R3). In view of the fact that N deposition is generally low in global forest ecosystems and rarely exceeding a maximum of 60 kg N ha$^{-1}$ yr$^{-1}$(Vet et al., 2014; Schwede et al., 2018), our results imply that separating the effects of N deposition and high-level N fertilization is necessary to avoid overestimate the effects of global N deposition. However, our new findings have never been considered by existing studies at a global scale (Liu and Greaver, 2009; Aronson and Helliker, 2010).

[Figure]

Figure R2. The mean difference (and 95% confidence intervals) of growing-season soil $CH_4$ flux to a) simulated N deposition (<60 kg $N^{-1}$ $yr^{-1}$), b) high-level N addition (> 60 kg $N^{-1}$ $yr^{-1}$) and c) all N treatments in boreal (BF), temperate (TemF), subtropical (STF), and tropical forest (TroF), respectively. The asterisk (*) indicates a significant effect (p<0.05).

[Figure]

Figure R3. The response ratio of growing-season soil $CH_4$ flux to simulated N deposition (LN, <60 kg $N^{-1}$ $yr^{-1}$), high-level N addition (HN, > 60 kg $N^{-1}$ $yr^{-1}$) and all N treatments (ALL) in boreal (BF), temperate (TemF) and subtropical forest (STF), respectively. Response ratio of insignificant effect (N treatments in tropical forest and LN in boreal forest) is not shown.

**Reference**

Aronson, E.L., Helliker, B.R. 2010. Methane flux in non‑wetland soils in response to nitrogen addition: a meta-analysis. Ecology, 91(11), 3242-3251.

Cleveland, C. C., Houlton, B. Z., Smith, W. K., Marklein, A. R., Reed, S. C., Parton, W., et al. 2013. Patterns of new versus recycled primary production in the terrestrial biosphere. Proceedings of the National Academy of Sciences, 110(31), 12733-12737.

Liu, L.L., and Greaver, T.L. 2009. A review of nitrogen enrichment effects on three biogenic GHGs: the $CO_2$ sink may be largely offset by stimulated $N_2O$ and $CH_4$ emission. Ecology Letters, 12, 1103–1117.

Schwede, D.B., Simpson, D., Tan, J., Fu, J., Dentener, F., Du, E., and De Vries, W.2018. Spatial variation of modelled total, dry and wet nitrogen deposition to forests at global scale. Environmental Pollution, 243, 1287-1301.

Vet, R., Artz, R.S., Carou, S., Shaw, M., Ro, C,U., Aas, W. et al. 2014.A global assessment of precipitation chemistry and deposition of sulfur, nitrogen, sea salt, base cations, organic acids, acidity and pH, and phosphorus, Atmospheric Environment, 93, 3–100.

Viechtbauer, W. 2010. Conducting meta-analyses in R with the metafor package. Journal of Statistical Software, 36(3), 1-48.

**Comments:** You did not mention any other criteria for inclusion or exclusion of studies. However, I think you should define what you consider a sufficiently large plot and a sufficiently wide buffer zone between treatments. Also, were all studies having true replicates? This kind of important information on the quality of studies is completely ignored in your manuscript.

**Reply:** Thanks for your suggestions. In the revised manuscript, we have updated the database by a) excluding experimental results from theses without peer-reviews, b) excluding experimental results from laboratory incubations, c) excluding an experimental study that didn't report values of variance/standardized error (no information for replicates), d) excluding results of experiments with plot area $< 10m^2$ or without buffer zones. We have also recorded information of replicates, plot area, buffer zone, and variance/standard deviation in our updated database. In the revised method section, we have included more detailed information for the criteria of data collection.

**Comments**: I found it very adventurous that you excluded the only peer-reviewed study conducted in tropical ecosystems (Veldkamp et al., 2013) with the argument that urea as an organic N form has 'limited implications

for the effects of N deposition', then cite Aronson and Helliker, (2010) as the source for this statement (page 3, line 20), and later lament that there are no studies conducted in tropical areas and then even extrapolate the results from subtropical forests to the tropics. -First of all, while urea is strictly spoken an organic N source, it is quickly hydrolysed ($NH_2CONH_2 + H_2O \rightarrow CO_2 + 2NH_3$) after application and the gaseous $NH_3$ reacts with water to form
5   ammonium ($NH_4^+$). Only on soils with a high pH there are significant losses through volatilization. -Second, in the paper by Aronson & Helliker (2010), I did not find any statement that urea has limited implications for the effects of N deposition. In contrast, they also analysed studies with urea additions. They found no difference between Urea and other pure N fertilizers. They also concluded that 'any conclusions of the effects of specific N species relative to others must be highly qualified, as the form of N that results may be quite different from that
10   added'. Therefore, to quote the Aronson & Helliker (2010) paper as the source why studies that add urea should be excluded is misleading. -Third, ignoring the only tropical study and later filling up the gap with studies from subtropical areas is very adventurous. Especially since the study conducted in the tropics did not following the hypothesized trend across biomes.

**Reply:** Thanks for your suggestions. We fully agree that it is not appropriate to exclude N addition experiments
15   using urea in tropical forests. In the paper by Aronson and Helliker (2010), the authors showed that the effects of N fertilization on soil $CH_4$ uptake did vary significantly with N forms (Figure R4). Specifically, the effects of urea and ammonium nitrate showed slight overlap, although the statistical analysis indicated no significant difference (Figure R4). This result motivated us to exclude experiments with urea additions in our previous manuscript, because a) N deposition mainly occurs in inorganic N forms, and b) urea might be less capable to
20   indicate the effects of N deposition. As suggested by you and other reviewers, we have updated our database by searching urea addition experiments in the literature in view of the fact that the effects of urea and ammonium nitrate were not significantly different. As a result, four experiments from tropical forest were included in our updated database (Veldkamp et al., 2013; Matson et al., 2016; Mori et al., 2017). Using a meta-analysis in R software with metaphor package (Viechtbauer, 2010), we show that simulated N deposition (<60 kg $N^{-1}$ $yr^{-1}$) and
25   high-level N addition (<60 kg $N^{-1}$ $yr^{-1}$) both had no significant effect on soil $CH_4$ uptake (Figure R2a&b). Overall, no significant effect of N additions on soil $CH_4$ uptake was found in tropical forest (Figure R2c). We have revised the manuscript accordingly.

[Figure]

Figure R4. Effects of different N forms on soil $CH_4$ uptake (Aronson and Helliker, 2010)

**Reference**:

Aronson, E.L., Helliker, B.R. 2010. Methane flux in non‐wetland soils in response to nitrogen addition: a meta-analysis. Ecology, 91(11), 3242-3251.

Matson, A. L. , Corre, M. D., Veldkamp, E. 2016. Canopy soil greenhouse gas dynamics in response to indirect fertilization across an elevation gradient of tropical montane forests. Biotropica, 49(2), 153-159.

Mori, T., Imai, N., Yokoyama, D., Mukai, M., Kitayama, K. 2017. Effects of selective logging and application of phosphorus and nitrogen on fluxes of $CO_2$, $CH_4$ and $N_2O$ in lowland tropical rainforests of borneo. Journal of Tropical Forest Science, 248-256.

Veldkamp, E., Koehler, B., Corre, M. D. 2013. Indications of nitrogen-limited methane uptake in tropical forest soils. Biogeosciences, 10(8), 5367-5379.

Viechtbauer, W. 2010. Conducting meta-analyses in R with the metafor package. Journal of Statistical Software, 36(3), 1-48.

**Comments**: The hypothesis to be tested (Page 2, line 21 and further) is weak and based on incomplete assumptions. You simply assume that N availability increases from boreal to tropical biomes, which is not true. If you read publications by Vitousek more carefully you will see that the main factor is not the biome but how heavily weathered soils are. More than half of the tropics is located on soils that are not heavily weathered (e.g. montane forests) and N availability is expected to be as low as other biomes where young, less weathered soils dominate.

**Reply:** Thanks for your comments. Nitrogen availability is internally driven by mineralization and externally from biological N fixation and N deposition. Recently, the role of N inputs from bedrock weathering is increasingly recognized, but this N source is only important when soil depth is shallow and N content is abundant

in the bedrock. Previous studies have shown that the sum of N inputs from biological N fixation and N deposition increase from boreal forest to tropical forests (Cleveland et al., 1999&2013; Du and De Vries, 2018). Moreover, the rate of N mineralization also shows an increase from boreal forest to tropical forest (Cleveland et al., 2013; Deng et al., 2018).  Overall, we think that the increase in N availability from boreal forest to tropical forest has been well recognized in literature.

**Reference**

Cleveland, C. C., Townsend, A. R., Schimel, D. S., Fisher, H., Howarth, R. W., Hedin, L. O., et al. 1999. Global patterns of terrestrial biological nitrogen ($N_2$) fixation in natural ecosystems. Global Biogeochemical Cycles, 13(2), 623-645.

Cleveland, C. C., Houlton, B. Z., Smith, W. K., Marklein, A. R., Reed, S. C., Parton, W., et al. 2013. Patterns of new versus recycled primary production in the terrestrial biosphere. Proceedings of the National Academy of Sciences, 110(31), 12733-12737.

Du, E., De Vries, W. 2018. Nitrogen-induced new net primary production and carbon sequestration in global forests, Environmental Pollution, 242, 1476–1487.

Deng, M., Liu, L., Jiang, L., Liu, W., Wang, X., Li, S., et al. 2018. Ecosystem scale trade-off in nitrogen acquisition pathways. Nature Ecology & Evolution, 2(11), 1724.

**Comments**: You group all forest ecosystems together and make no difference between natural forest and plantations or managed forests. Tree plantations typically have significant growth rates and are almost always N limited, also in tropical and subtropical conditions. Ignoring this may lead to wrong conclusions.

**Reply:** This is a good point. We have checked our database and include information of natural forest or plantations. We found that only 10 of 28 forests (1/7 boreal, 3/7 temperate and 6/10 subtropical) were plantations. This doesn't allow us to test the difference between plantations and natural forests for each forest biome due to uneven small sample size. As the database is the most updated from literature, our results represent the best knowledge we can derive at this stage. In the revised manuscript, we have discussed the uncertainties possibly due to the different N status of natural forest and plantations. We have also labelled natural forests and plantations in the revised summary table for the database.

**Comments**: In summary, this synthesis is poorly conducted. There are studies included that are not peer-reviewed and there were no quality criteria for the studies that were included. The hypothesis is weak and based on incomplete assumptions. The distinction between 'high level' and 'low level' N addition is arbitrary and the

background N deposition is ignored. Finally, I could not find any objective reason why studies where urea was added were excluded. Given these weaknesses of this synthesis I strongly doubt the validity of the conclusions and I recommend not to publish this manuscript in Biogeosciences.

**Reply**: Thanks again for your helpful comments. First, we have updated the manuscript by a) excluding experimental results from master thesis, b) excluding experimental results from laboratory incubations, c) excluding an experimental study that didn't report values of variance/standardized error, d) excluding results of experiments with plot area $< 10m^2$ or without buffer zones, and e) including more results of field experiments using urea additions. More detailed information for data collection has also been included in the revised section on data and method. Second, we have included more detailed evidence from literature for the gradients of N availability across forest biomes, both internally driven by mineralization and externally from biological N fixation and N deposition. This underpins our hypothesis that the effect of N additions likely varies across forest biomes due to a shift in N availability. Third, the threshold of maximum N deposition versus high-level N addition is defined based on a global assessment of N deposition by the World Meteorological Organization (WMO) Global Atmosphere Watch programme (GAW), which shows a range of N deposition in various regions of the world, from 1~62.25 kg N ha$^{-1}$ yr$^{-1}$. We thus used 60 kg N ha$^{-1}$ yr$^{-1}$ as a threshold for a possible maximum of N deposition, in order to distinguish it from N fertilization with extremely high levels of N additions. Finally, we have updated our database by including urea addition experiments and conducted a reanalysis using meta-analysis. Overall, we believe that the main concerns have been well addressed in the revised manuscript.

---

## Author Comment (AC6) · 20 Apr 2019

**Reply to the comments by reviewer #2**

Enzai Du[1,2*], Nan Xia[2], Wim de Vries[3,4]

[1]State Key Laboratory of Earth Surface Processes and Resource Ecology, Faculty of Geographical Science, Beijing Normal University, Beijing100875, China

5 [2]School of Natural Resources, Faculty of Geographical Science, Beijing Normal University, Beijing100875, China

[3]Wageningen University and Research, Environmental Research, PO Box 47, NL-6700 AA Wageningen, the Netherlands

[4]Wageningen University and Research, Environmental Systems Analysis Group, PO Box 47, NL-6700 AA Wageningen, the Netherlands

10 *Correspondence to*: Enzai Du (enzaidu@bnu.edu.cn)

**Comments**: This manuscript reports on a synthesis or meta-study of studies that report the effects of N additions on soil CH4 fluxes. The authors did a literature search using several sources, calculated a response variable, stratified results according to dose and biome, and conclude that low-level N addition increased $CH_4$ uptake in boreal forest while decreases occurred at high N addition levels and in all other biomes. A meta-study is only as

15 good as the sources where the information comes from, and therefore studies used in such a meta-study should be generally accessible and peer-reviewed. Unfortunately, several of the studies used in this meta-study did not meet these criteria. I find it problematic that apart from ISI Web of Science, also Google Scholar and China National Knowledge Infrastructure are used to search for literature. To my knowledge, these two latter sources also list reports that are not peer-reviewed. I strongly suggest to ONLY use ISI as a literature source, because here only

20 peer-reviewed sources are listed. Studies that are not peer reviewed should not be included in a synthesis. In your meta-study I suggest to exclude the following studies listed in the supplement: - two of the studies listed are a MSc thesis (Wang, 2012, Pan, 2013). Such a thesis is not considered peer-reviewed, please exclude them. -all studies that involved incubations, instead of field measurements. For example, the study by Chen et al. 2017 mentions that they did laboratory incubations. You even state in Page 3, line 16 that you only included studies

25 that used closed static chamber technology. Apparently that is not true. There may be more studies with incubations, I did not check them all. - Please, exclude studies published in Chinese (or other non-English publications) with only an English abstract. I do not consider such studies as generally accessible. For example, the study by Hu et al., 2011, is only accessible in Chinese and there may be more in the list.

**Reply:** Thanks for your comments, which have helped us to improve the manuscript substantially. We fully agree

30 that the quality of data sources is very important.  In the revised manuscript, we have updated the database by a) excluding experimental results from theses without peer-review (Wang, 2012; Pan, 2013), b) excluding experimental results from laboratory incubations (only Chen et al. 2017), c) excluding an experimental study that

didn't report values of variance/standardized error (Steudler et al., 1989), d) excluding results of experiments with plot area < 10m$^2$ or without buffer zones, and e) including more results of field experiments using urea additions. We further beg your understanding to include reports published in fully peer-reviewed Chinese journals because these publications are generally of good quality in science. Overall, we updated our database based on the criteria above and the current database includes results of 28 experiments across 22 sites (Figure R1).

[Figure]

Figure R1. Geographical distribution of 28 forests receiving experimental nitrogen additions. Green shadows indicate the distribution of global forest.

**Reference**

Pan, D.R. 2013. Study on greenhouse gas emission for grassland soil below different forest soils under precipitation reduction and Nitrogen deposition in Shennongjia mountain. Gansu Agricultural University, Master thesis.

Steudler, P.A., Bowden, R.D., Melillo, J.M., Aber, J.D. 1989. Influence of nitrogen fertilization on methane uptake in temperate forest soils. Nature, 341(6240), 314-316.

Wang, R.N. 2012. Effects of simulated atmospheric nitrogen deposition on the exchange fluxes of greenhouse gases in the temperate forest soil. Beijing Forestry University, Master thesis

**Comments**: You use the 60 kg N ha$^{-1}$ yr$^{-1}$ as an arbitrary cut-off between 'low' and 'high' level N addition. Did you calculate also the background N-deposition in the N additions? For example, the study by Li et al., 2015, mentions that there is a background N deposition of more than 30 kg N ha$^{-1}$ yr$^{-1}$, while the treatment is 40 kg ha$^{-1}$ yr$^{-1}$. Together this would be more than 60 kg N ha$^{-1}$ yr$^{-1}$ and the 40 kg treatment should be grouped as 'high' level

N addition. I suspect that is not how you did this and it just illustrates how arbitrary the choice of 60 kg N ha$^{-1}$ yr$^{-1}$ is.

**Reply:** Thanks for your comments. We defined the threshold of N deposition versus high-level N addition based on a global assessment of N deposition by the World Meteorological Organization (WMO) Global Atmosphere Watch programme (GAW) (Vet et al., 2014). This assessment shows a range of N deposition in various regions of the world, from 1~62 kg N ha$^{-1}$ yr$^{-1}$. Specifically, the maximum level of N deposition occurs in eastern and southern China (Vet et al., 2014). We thus used 60 kg N ha$^{-1}$ yr$^{-1}$ as a threshold for a possible maximum of N deposition, in order to distinguish it from N fertilization with extremely high levels of N additions. In the revised manuscript, we have conducted a meta-analysis (Viechtbauer, 2010) by using background N deposition as a moderator. We used a mean difference (Flux$_{treatment}$-Flux$_{control}$, the difference of mean growing-season soil $CH_4$ fluxes between the treatment plots and control plots) as the effect size to evaluate the effect of N additions. Including background N deposition to N addition is normally not done as the N addition effect is compared to a control receiving the same deposition and addition may lead to erroneous interpretations (e.g Hetterlingh et al., 2015). The standard approach is to assess the impact of background deposition as a moderator of the N addition effects (see e.g. Granath et al., 2014; Midolo et al., 2018). We did so and our results indicate that background N deposition has a significant influence on the effect of N additions (p=0.0006). However, since average N deposition varied significantly across global forest biomes, showing a trend from subtropical forest (14.6 kg N ha$^{-1}$ yr$^{-1}$) > temperate forest (7.3 kg N ha$^{-1}$ yr$^{-1}$) and tropical forest (7.2 kg N ha$^{-1}$ yr$^{-1}$) > boreal forest (1.2 kg N ha$^{-1}$ yr$^{-1}$), this hinders us to separate the effects of background N deposition from the effect of forest biomes. We have revised the manuscript to avoid any misunderstanding.

**Reference**

Aronson, E.L., Helliker, B.R. 2010. Methane flux in non‐wetland soils in response to nitrogen addition: a meta-analysis. Ecology, 91(11), 3242-3251.

Cleveland, C. C., Houlton, B. Z., Smith, W. K., Marklein, A. R., Reed, S. C., Parton, W., et al. 2013. Patterns of new versus recycled primary production in the terrestrial biosphere. Proceedings of the National Academy of Sciences, 110(31), 12733-12737.

Granath, G., Limpens, J., Posch, M., Mücher, S., de Vries, W.. 2014. Spatio-temporal trends of nitrogen deposition and climate effects on Sphagnum productivity in European peatlands. Environmental Pollution 187, 73-80.

Hettelingh, J-P., Stevens, C., Posch, M., Bobbink, R.,  de Vries, W. 2015. Assessing the impacts of nitrogen deposition on plant species richness in Europe. In W. de Vries, J-P. Hettelingh & M. Posch (eds) Critical Loads and Dynamic Risk Assessments: Nitrogen, Acidity and Metals in Terrestrial and Aquatic Ecosystems, Environmental Pollution Volume 25, Springer ISSN 1566-0745, 573-586.

Liu, L.L., and Greaver, T.L. 2009. A review of nitrogen enrichment effects on three biogenic GHGs: the $CO_2$ sink may be largely offset by stimulated $N_2O$ and $CH_4$ emission. Ecology Letters, 12, 1103–1117.

Midolo, G., Alkemade, R., Schipper, A.M., Benítez-López, A., Perring, M.P., de Vries, W. 2018. Impacts of nitrogen addition on plant species richness and abundance: A global meta-analysis. Global Ecology and Biogeography, 28(3), 398-413.

Schwede, D.B., Simpson, D., Tan, J., Fu, J., Dentener, F., Du, E., and De Vries, W.2018. Spatial variation of modelled total, dry and wet nitrogen deposition to forests at global scale. Environmental Pollution, 243, 1287-1301.

Vet, R., Artz, R.S., Carou, S., Shaw, M., Ro, C,U., Aas, W. et al. 2014.A global assessment of precipitation chemistry and deposition of sulfur, nitrogen, sea salt, base cations, organic acids, acidity and pH, and phosphorus, Atmospheric Environment, 93, 3–100.

Viechtbauer, W. 2010. Conducting meta-analyses in R with the metafor package. Journal of Statistical Software, 36(3), 1-48.

**Comments:** You did not mention any other criteria for inclusion or exclusion of studies. However, I think you should define what you consider a sufficiently large plot and a sufficiently wide buffer zone between treatments. Also, were all studies having true replicates? This kind of important information on the quality of studies is completely ignored in your manuscript.

**Reply:** Thanks for your suggestions. In the revised manuscript, we have updated the database by a) excluding experimental results from theses without peer-reviews, b) excluding experimental results from laboratory incubations, c) excluding an experimental study that didn't report values of standarde deviation/standardized error (no information for replicates), d) excluding results of experiments with plot area $< 10m^2$ or without buffer zones. We have also recorded information of replicates, plot area, buffer zone, and variance/standard deviation in our updated database. In the revised method section, we have included more detailed information for the criteria of data collection.

**Comments**: I found it very adventurous that you excluded the only peer-reviewed study conducted in tropical ecosystems (Veldkamp et al., 2013) with the argument that urea as an organic N form has 'limited implications for the effects of N deposition', then cite Aronson and Helliker, (2010) as the source for this statement (page 3, line 20), and later lament that there are no studies conducted in tropical areas and then even extrapolate the results from subtropical forests to the tropics. -First of all, while urea is strictly spoken an organic N source, it is quickly hydrolysed ($NH_2CONH_2 + H_2O \rightarrow CO_2 + 2NH_3$) after application and the gaseous $NH_3$ reacts with water to form ammonium ($NH_4^+$). Only on soils with a high pH there are significant losses through volatilization. -Second, in the paper by Aronson & Helliker (2010), I did not find any statement that urea has limited implications for the effects of N deposition. In contrast, they also analysed studies with urea additions. They found no difference between Urea and other pure N fertilizers. They also concluded that 'any conclusions of the effects of specific N species relative to others must be highly qualified, as the form of N that results may be quite different from that added'. Therefore, to quote the Aronson & Helliker (2010) paper as the source why studies that add urea should be excluded is misleading. -Third, ignoring the only tropical study and later filling up the gap with studies from subtropical areas is very adventurous. Especially since the study conducted in the tropics did not following the hypothesized trend across biomes.

**Reply:** Thanks for your suggestions. We fully agree that it is not appropriate to exclude N addition experiments using urea in tropical forests. In the paper by Aronson and Helliker (2010), the authors showed that the effects of N fertilization on soil $CH_4$ uptake did vary significantly with N forms (Figure R2). Specifically, the effects of urea and ammonium nitrate showed slight overlap, although the statistical analysis indicated no significant difference (Figure R2). This result motivated us to exclude experiments with urea additions in our previous manuscript, because a) N deposition mainly occurs in inorganic N forms, and b) urea might be less capable to indicate the effects of N deposition. As suggested by you and other reviewers, we have updated our database by searching urea addition experiments in the literature in view of the fact that the effects of urea and ammonium nitrate were not significantly different. As a result, four experiments from tropical forest were included in our updated database (Veldkamp et al., 2013; Matson et al., 2016; Mori et al., 2017).

Using a meta-analysis in R software with metaphor package (Viechtbauer, 2010), we show that low-level N addition ($<60$ kg $N^{-1}$ $yr^{-1}$, in the range comparable to N deposition) and high-level N fertilization ($>60$ kg $N^{-1}$ $yr^{-1}$) both had no significant effect on soil $CH_4$ uptake (Figure R3a&b). Overall, no significant effect of N additions on soil $CH_4$ uptake was found in tropical forest (Figure R3c). We have revised the manuscript accordingly.

[Figure]

Figure R2. Effects of different N forms on soil CH$_4$ uptake (Aronson and Helliker, 2010)

[Figure]

Figure R3. The mean difference (and 95% confidence intervals) of growing-season soil CH$_4$ flux to a) low-level N addition (<60 kg N$^{-1}$ yr$^{-1}$), b) high-level N fertilization (> 60 kg N$^{-1}$ yr$^{-1}$) and c) all N treatments in boreal (BF), temperate (TemF), subtropical (STF), and tropical forest (TroF), respectively. The asterisk (*) indicates a significant effect (p<0.05).

**Reference**:

Aronson, E.L., Helliker, B.R. 2010. Methane flux in non‐wetland soils in response to nitrogen addition: a meta-analysis. Ecology, 91(11), 3242-3251.

Matson, A. L. , Corre, M. D., Veldkamp, E. 2016. Canopy soil greenhouse gas dynamics in response to indirect fertilization across an elevation gradient of tropical montane forests. Biotropica, 49(2), 153-159.

Mori, T., Imai, N., Yokoyama, D., Mukai, M., Kitayama, K. 2017. Effects of selective logging and application of phosphorus and nitrogen on fluxes of $CO_2$, $CH_4$ and $N_2O$ in lowland tropical rainforests of borneo. Journal of Tropical Forest Science, 248-256.

Veldkamp, E., Koehler, B., Corre, M. D. 2013. Indications of nitrogen-limited methane uptake in tropical forest soils. Biogeosciences, 10(8), 5367-5379.

Viechtbauer, W. 2010. Conducting meta-analyses in R with the metafor package. Journal of Statistical Software, 36(3), 1-48.

**Comments**: The hypothesis to be tested (Page 2, line 21 and further) is weak and based on incomplete assumptions. You simply assume that N availability increases from boreal to tropical biomes, which is not true. If you read publications by Vitousek more carefully you will see that the main factor is not the biome but how heavily weathered soils are. More than half of the tropics is located on soils that are not heavily weathered (e.g. montane forests) and N availability is expected to be as low as other biomes where young, less weathered soils dominate.

**Reply:** Thanks for your comments. Nitrogen availability is internally driven by mineralization and externally from biological N fixation and N deposition. Recently, the role of N inputs from bedrock weathering is increasingly recognized, but the role of this N source for uptake from shallow soil depth is very limited. Previous studies have shown that the sum of N inputs from biological N fixation and N deposition dominate the external N input and these inputs increase from boreal forest to tropical forests (Cleveland et al., 1999&2013; Du and De Vries, 2018). Moreover, the rate of N mineralization also shows an increase from boreal forest to tropical forest (Cleveland et al., 2013; Deng et al., 2018). Overall, we think that the increase in N availability from boreal forest to tropical forest has been well recognized in literature.

**Reference**

Cleveland, C. C., Townsend, A. R., Schimel, D. S., Fisher, H., Howarth, R. W., Hedin, L. O., et al. 1999. Global patterns of terrestrial biological nitrogen ($N_2$) fixation in natural ecosystems. Global Biogeochemical Cycles, 13(2), 623-645.

Cleveland, C. C., Houlton, B. Z., Smith, W. K., Marklein, A. R., Reed, S. C., Parton, W., et al. 2013. Patterns of new versus recycled primary production in the terrestrial biosphere. Proceedings of the National Academy of Sciences, 110(31), 12733-12737.

Du, E., De Vries, W. 2018. Nitrogen-induced new net primary production and carbon sequestration in global forests, Environmental Pollution, 242, 1476–1487.

Deng, M., Liu, L., Jiang, L., Liu, W., Wang, X., Li, S., et al. 2018. Ecosystem scale trade-off in nitrogen acquisition pathways. Nature Ecology & Evolution, 2(11), 1724.

**Comments**: You group all forest ecosystems together and make no difference between natural forest and plantations or managed forests. Tree plantations typically have significant growth rates and are almost always N limited, also in tropical and subtropical conditions. Ignoring this may lead to wrong conclusions.

**Reply:** This is a good point. We have checked our database and include information of natural forest or plantations. We found that only 10 of 28 forests (1/7 boreal, 3/7 temperate and 6/10 subtropical) were plantations. This doesn't allow us to test the difference between plantations and natural forests for each forest biome due to uneven small sample size. As the database is the most updated from literature, our results represent the best knowledge we can derive at this stage. In the revised manuscript, we have discussed the uncertainties possibly due to the different N status of natural forest and plantations. We have also labelled natural forests and plantations in the revised summary table for the database.

**Comments**: In summary, this synthesis is poorly conducted. There are studies included that are not peer-reviewed and there were no quality criteria for the studies that were included. The hypothesis is weak and based on incomplete assumptions. The distinction between 'high level' and 'low level' N addition is arbitrary and the background N deposition is ignored. Finally, I could not find any objective reason why studies where urea was added were excluded. Given these weaknesses of this synthesis I strongly doubt the validity of the conclusions and I recommend not to publish this manuscript in Biogeosciences.

**Reply**: Thanks again for your helpful comments. First, we have updated the manuscript by a) excluding experimental results from theses, b) excluding experimental results from laboratory incubations, c) excluding an experimental study that didn't report values of standard deviations/standardized error, d) excluding results of experiments with plot area $< 10m^2$ or without buffer zones, and e) including more results of field experiments using urea additions. More detailed information for standard data collection has also been included in the revised section on data and method. Second, we have included more detailed evidence from literature for the gradients of N availability across forest biomes, both internally driven by mineralization and externally from biological N fixation and N deposition. This underpins our hypothesis that the effect of N additions likely varies across forest biomes due to a shift in N availability. Third, the threshold of maximum N deposition versus high-level N addition is defined based on a global assessment of N deposition by the World Meteorological Organization (WMO) Global Atmosphere Watch programme (GAW) (Vet et al., 2014), which shows a range of N deposition

in various regions of the world, from 1~62 kg N ha$^{-1}$ yr$^{-1}$. We thus used 60 kg N ha$^{-1}$ yr$^{-1}$ as a threshold for a possible maximum of N deposition, in order to distinguish it from N fertilization with extremely high levels of N additions. Fourth, we have updated our database by including urea addition experiments and conducted a reanalysis using meta-analysis. Finally, we included background N deposition as a moderator of N addition

5  effects. The renewed analysis indicates that only temperate forest shows a significant response ratio to low-level N addition, while high-level N fertilization results in significant response ratios across boreal, temperate and subtropical forests (Fig. R4). In view of the fact that N deposition is generally low in global forest ecosystems and rarely exceeding a maximum of 60 kg N ha$^{-1}$ yr$^{-1}$(Vet et al., 2014; Schwede et al., 2018), our results imply that separating the effects of N deposition and high-level N fertilization is necessary to avoid overestimate the

10  effects of global N deposition. However, this has never been considered by existing meta-analyses at a global scale (Liu and Greaver, 2009; Aronson and Helliker, 2010). Overall, we believe that the main concerns have been well addressed in the revised manuscript.

[Figure]

Figure R4. The response ratio of growing-season soil CH$_4$ flux to low-level N addition (LN, <60 kg N$^{-1}$ yr$^{-1}$),

15  high-level N fertilization (HN, > 60 kg N$^{-1}$ yr$^{-1}$) and all N treatments (ALL) in boreal (BF), temperate (TemF) and subtropical forest (STF), respectively. Response ratio of insignificant effect (N treatments in tropical forest and LN in boreal and subtropical forest) is not shown.

**Reference**

Aronson, E.L., Helliker, B.R. 2010. Methane flux in non‑wetland soils in response to nitrogen addition: a meta-

20     analysis. Ecology, 91(11), 3242-3251.

Liu, L.L., and Greaver, T.L. 2009. A review of nitrogen enrichment effects on three biogenic GHGs: the $CO_2$ sink may be largely offset by stimulated $N_2O$ and $CH_4$ emission. Ecology Letters, 12, 1103–1117.

Schwede, D.B., Simpson, D., Tan, J., Fu, J., Dentener, F., Du, E., and De Vries, W.2018. Spatial variation of modelled total, dry and wet nitrogen deposition to forests at global scale. Environmental Pollution, 243, 1287-1301.

Vet, R., Artz, R.S., Carou, S., Shaw, M., Ro, C,U., Aas, W. et al. 2014.A global assessment of precipitation chemistry and deposition of sulfur, nitrogen, sea salt, base cations, organic acids, acidity and pH, and phosphorus, Atmospheric Environment, 93, 3–100.